# Dynamics of non-self-similar earthquakes illuminated by a controlled fault asperity

Kurama Okubo [1] ✉, Futoshi Yamashita [1,3] & Eiichi Fukuyama [1,2,3]

Most ordinary earthquakes follow self-similar scaling, where source duration scales with the cube root of seismic moment. However, some earthquake clusters show non-self-similar scaling, in which source duration remains nearly constant regardless of seismic moment. Their source mechanisms, previously proposed to involve fixed source dimensions with variable stress drop or accelerating rupture velocity, are not fully validated due to uncertainties in estimating source properties, often caused by observational biases such as path effects. Here, we present a dynamic rupture model for non-self-similar earthquakes based on laboratory experiments conducted on a meter-scale fault with size- and shape-controlled gouge patch sources. By carefully applying corrections for instrumental response, sensor coupling, and attenuation to acoustic emission waveforms, we reliably constrain the source parameters of gouge patch events and identify non-self-similar scaling across magnitudes from $M_w$ -7.3 to -6.0. We further develop a dynamic rupture model that quantitatively explains the observed source parameters by incorporating a fixed source-patch size, variable stress drop within the patch, and self-healing friction. This modeling framework complements previously proposed models and expands the range of tectonic conditions under which non-self-similar earthquakes may occur.

The scaling of ordinary earthquakes brings insight into the fundamentals of earthquake rupture dynamics[1–6]. Classic source models based on a shear crack (e.g.[7]), assuming constant stress drop and rupture velocity while allowing for variable source dimensions, predict self-similar scaling, where seismic moment and source duration follow a cubic relationship (e.g.[8,9]). However, deviations from this scaling have been observed in some earthquake clusters, showing a nearly constant source duration regardless of variations in seismic moment (e.g.[10,11]).

This scaling relationship, referred to as non-self-similar scaling, was first reported in earthquake clusters at Parkfield by Harrington and Brodsky[10] and later observed beneath the Chelungpu thrust fault by Lin et al.[11]. Nakajima and Hasegawa[12] compiled observations of repeating earthquakes that exhibit non-self-similar scaling across various tectonic settings, including the continental crust, intraslab regions, and

plate boundaries beneath the Japanese Islands. In addition, such scaling behavior has been observed in low-frequency earthquakes[13,14], fluid-induced seismicity[15,16], and foreshock sequences[17].

In this study, we define non-self-similar earthquakes as clusters of events generated by a non-self-similar source process, as illustrated in Fig. 1a. We focus on clusters in which events originate from a common source location and show highly coherent waveform phases but variable amplitudes. Clusters that exhibit deviations from self-similar behavior less pronounced than those considered in the present analysis (e.g.[18]) are not examined.

Under this framework, non-self-similar earthquakes that share a common source location are treated as a subset of repeating earthquakes (Supplementary Fig. S1), without considering recurrence intervals as a classification criterion. Consequently, repeating earthquakes include non-self-similar clusters and the complementary

[1]National Research Institute for Earth Science and Disaster Resilience (NIED), Tsukuba, Japan. [2]Department of Civil and Earth Resources Engineering, Kyoto University, Kyoto, Japan. [3]These authors contributed equally: Futoshi Yamashita, Eiichi Fukuyama. ✉e-mail: kokubo@bosai.go.jp

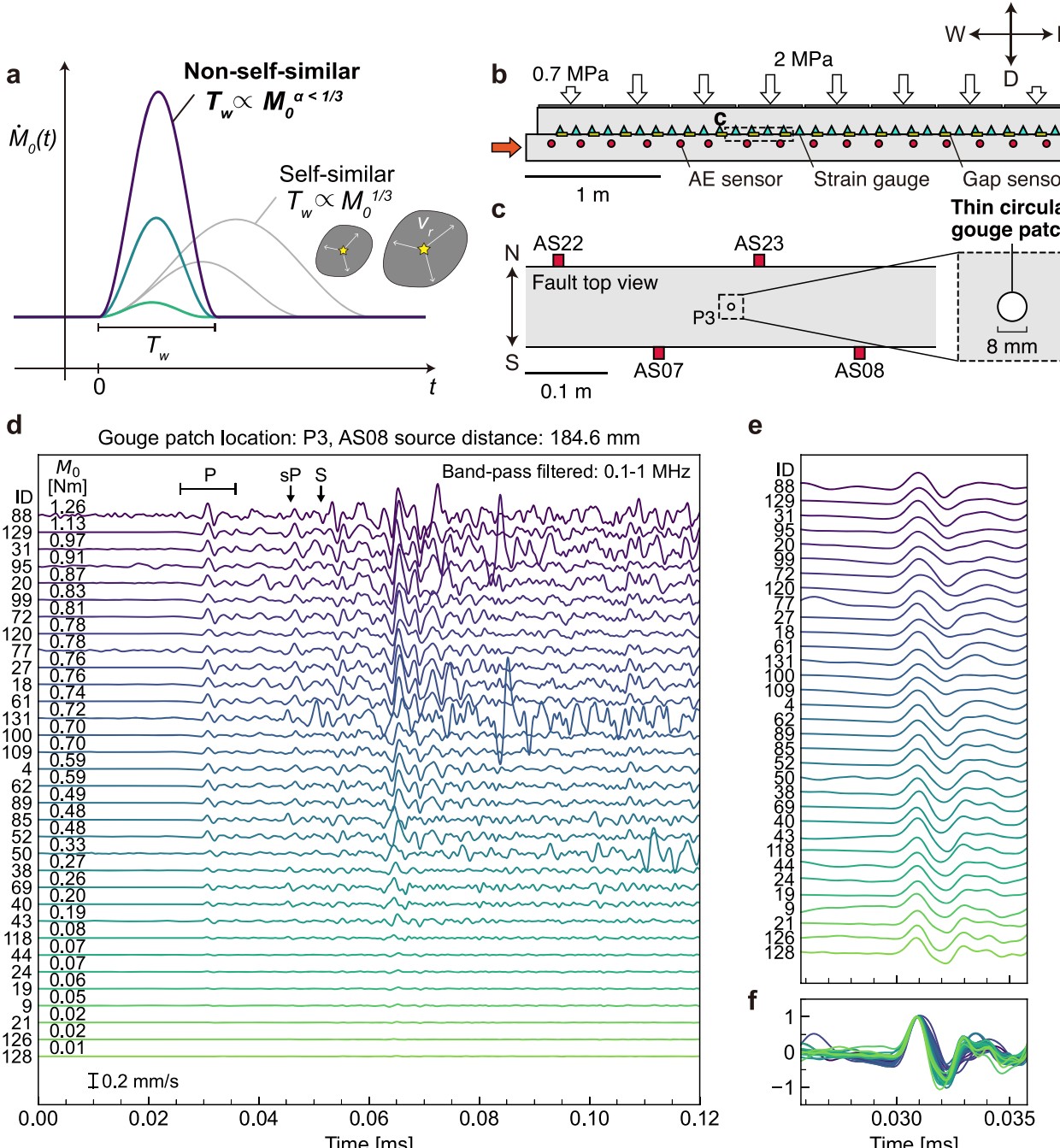

**Fig. 1 | Generation of the non-self-similar earthquakes by the controlled fault asperity. a** Schematic illustrating the source process for self-similar and non-self-similar earthquakes. **b** Side view of the large-scale biaxial rock friction apparatus with 4 m long rock specimens. **c** Top view of the simulated fault showing the location of the gouge patch (GP) at P3. The four nearby acoustic emission (AE) sensors used in the analysis are also indicated. **d** Collection of AE waveforms recorded by AS08 for the foreshocks and aftershocks generated by P3. The instrumental response is corrected, and a 0.1-1 MHz two-way band-pass filter is applied to the waveforms. The synthetic travel times are annotated at the top, where *sP* indicates the sP-converted wave reflected from the side surface of the rock specimen. Events are numbered chronologically based on their onset time. The seismic moment ($M_0$) is estimated by fitting the synthetic source time function (STF; "Methods"), and line color brightness corresponds to $M_0$. Out of 44 events, 33 that exceed the quality thresholds (Methods) are shown. **e** Normalized P-waveforms corresponding to the annotated window in (**d**) aligned by the positive peak of the P pulse. Line colors match those in (**d**). **f** Superimposed P-waveforms normalized by amplitude, shown within the same time window as in (**e**).

clusters composed of events with nearly identical waveforms in both phase and amplitude. This classification follows common practice in observational seismology, where events with coherent waveforms are typically classified as repeating earthquakes even when modest amplitude variations are present (e.g.[19]). Our framework aligns with

established observational conventions while explicitly distinguishing non-self-similar behavior within repeating earthquake clusters.

Several source models associated with isolated asperity patches have been proposed to explain the mechanisms of non-self-similar earthquakes[12,20–22]. However, constraining these models remains

challenging due to observational biases, such as attenuation effects (e.g.[23]), and the limited frequency range of the instruments, which affect the estimation of source properties. While previous studies have carefully accounted for these biases, further refinement of source characteristics is necessary to better understand the origins of non-self-similarity and its underlying source mechanisms.

Here, to address these challenges, we investigate earthquakes generated by size- and shape-controlled asperities on a meter-scale laboratory fault. The precise calibration of measurement systems and correction for attenuation effects enable a more accurate estimation of scaling laws compared to natural observations. Using well-constrained kinematic source parameters and the known geometry of the asperity patch, we constrain a dynamic rupture model and identify key mechanisms that reproduce non-self-similar scaling.

## Results

### Experimental setup

A large-scale biaxial rock friction apparatus consists of a pair of vertically stacked rock specimens (Fig. 1b), with a simulated fault having a nominal contact area of $4.0 \times 0.1\,\mathrm{m}^2$. Normal loading is applied via eight flat jacks inserted between the upper rock specimen and the outer frame, controlling the spatial distribution of the normal stress, which is set to 2.0 MPa in the center of the fault and 0.7 MPa at both edges. The decreased normal loading at the edges promotes preslip propagation from the fault edges prior to the stick-slip events.

To monitor the evolution of slip and stress along the fault, strain gauges were installed on the side surface of the upper specimen, 10 mm above the fault, while gap sensors were placed across the fault on the side surface to measure the relative displacement. Furthermore, acoustic emission (AE) sensors were installed on the side surface of the lower specimen, 70 mm below the fault, to record AE waveform data continuously. The specifications of these measurement systems are detailed in the Methods section, and the configuration of the sensor array is illustrated in Supplementary Fig. S2.

To control the size, shape, and location of the sources, we placed thin circular gouge patches (GPs) on the middle of the fault as fault asperities (Fig. 1c and Supplementary Fig. S3a, b and Methods for GP setup). We selected the size of the GP to be 8 mm in diameter for the following reasons: (i) the expected corner frequency associated with a seismic source of this size is approximately 300 kHz based on the classical shear crack model (e.g.[7]), which lies within the measurement range of our AE sensors (see Methods); (ii) the patch size must exceed the critical nucleation size, which we estimated to be on the order of a few millimeters based on the normal stress on the patch inferred from pressure-sensitive film measurements (see "Methods"), together with assumed frictional parameters of the slip-weakening law; and (iii) the ratio of the patch size to the overall simulated fault dimension is sufficiently small to approximate a buried source in an effectively infinite medium, consistent with natural settings.

The gouge was prepared by pulverizing a block of the same rock type as the host rock specimens, metagabbro, to mimic the natural heterogeneity and the frictional properties produced by wear on the laboratory fault. The GP region has a slightly higher topographic height than the surrounding fault surface. This topographic offset leads to a concentration of normal stress onto the GP when the upper rock specimen is stacked, which strengthens the frictional coupling. We located seven GPs at 500 mm intervals, starting 750 mm from the western edge of the fault (Supplementary Fig. S2c, d), which was designed to avoid the interference of the radiated waves between the neighboring GPs.

### Non-self-similarity of GP-generated events

We observed 56 stick-slip events that ruptured the entire simulated fault during a single experimental run. The average slip per event was $31.3 \pm 4.0\,\mu\mathrm{m}$, and the total slip was 2.1 mm during the run. We then detected 107 GP-generated seismic events in total as the fore- or aftershocks during the evolution of preslip and afterslip. The third GP from the west, labeled P3, was the most activated one among the seven GPs (Supplementary Fig. S4a). The difference in the activity of the GPs is most likely caused by the non-uniform normal stress distribution on the fault, as inferred from the near-fault strain measurements (Supplementary Fig. S4b).

The AE waveforms generated by P3 during individual events, after correction for the sensor instrumental response (Methods), show a high degree of phase correlation (Fig. 1d and Supplementary Fig. S5). Although slight dissimilarities exist, likely caused by variations in rupture nucleation locations and scatterers, the consistent P, S, and side-surface reflected waves indicate that these events originated from the same source patch. This contrasts with the case of Ellsworth and Bulut[24], where waveform dissimilarity was critical for identifying multiple nearby source patches.

While the seismic moment ($M_0$; see Methods for its evaluation) spans nearly two orders of magnitude, the normalized P-wave pulses show high correlation, even among smaller-magnitude events (Fig. 1e, f), consistent with non-self-similar scaling. We performed comprehensive calibration and attenuation corrections to refine the scaling relationship.

The GP-generated foreshocks occurred after the passage of the propagation front of preslips nucleated from the edge of the fault (Fig. 2a). The slip measured at the side surface of the fault was accumulated before the onset of GP events (Fig. 2b), which would indicate that the GP events were triggered due to the loading by the aseismic slip around the patch, as proposed by the early laboratory studies[25,26] and the observations (e.g.[27]). The zoomed AE waveforms show that the P-wave arrival times and their polarities match the location of the GP and the slip direction parallel to the shear loading (Fig. 2c). The aftershocks were generated during the afterslip (Supplementary Fig. S6), which could be caused by the transient healing of the frictional strength on the GP.

To accurately evaluate the source parameters and scaling law of the GP events, it is essential to correct for the amplitude and frequency response of the AE sensors as well as path effects[28–30]. To achieve this, we first performed a sensor calibration by modeling their response using poles and zeros, which allows for a similar process of instrumental response removal as with seismometers (Methods). Although precisely characterizing the sensor response remains challenging, this approach improves the robustness of the correction to evaluate the velocity waveforms from the AE sensor outputs. In addition, we calibrated the sensor coupling factor for each sensor installed on the rock specimen using a ball-drop impact test ("Methods"). These careful sensor calibrations enable the quantitative evaluation of the source parameters on the GP events using the AE waveforms.

The attenuation structure of the crust is complex, hindering accurate correction of the observed waveforms in natural settings. In contrast, the structure of the laboratory rock specimen is relatively simpler, allowing us to estimate a plausible frequency-dependent attenuation model, $Q_p^{-1}(f)$ ("Methods" and Supplementary Fig. S7). We applied this model to the recorded P-velocity waveforms by deconvolving the $Q_p^{-1}(f)$ and computed the P-wave displacement pulses by numerically integrating the corrected velocity waveforms.

We evaluated the source parameters by fitting the synthetic source time function (STF) in the time domain to the far-field P-wave displacement pulses after converting it to the moment rate, accounting for the radiation pattern and the reflection coefficient ("Methods" and Fig. 3a, b). While various metrics have been used to quantify the source characteristics, such as the half-maximum pulse width of the STF[10,11] or the spectral ratio[12,15–17], we chose to fit the synthetic STF for our analysis. This approach allows the quality of the source parameter estimates to be assessed through the residuals between the observed and best-fit STFs ("Methods"). The rupture directivity was not

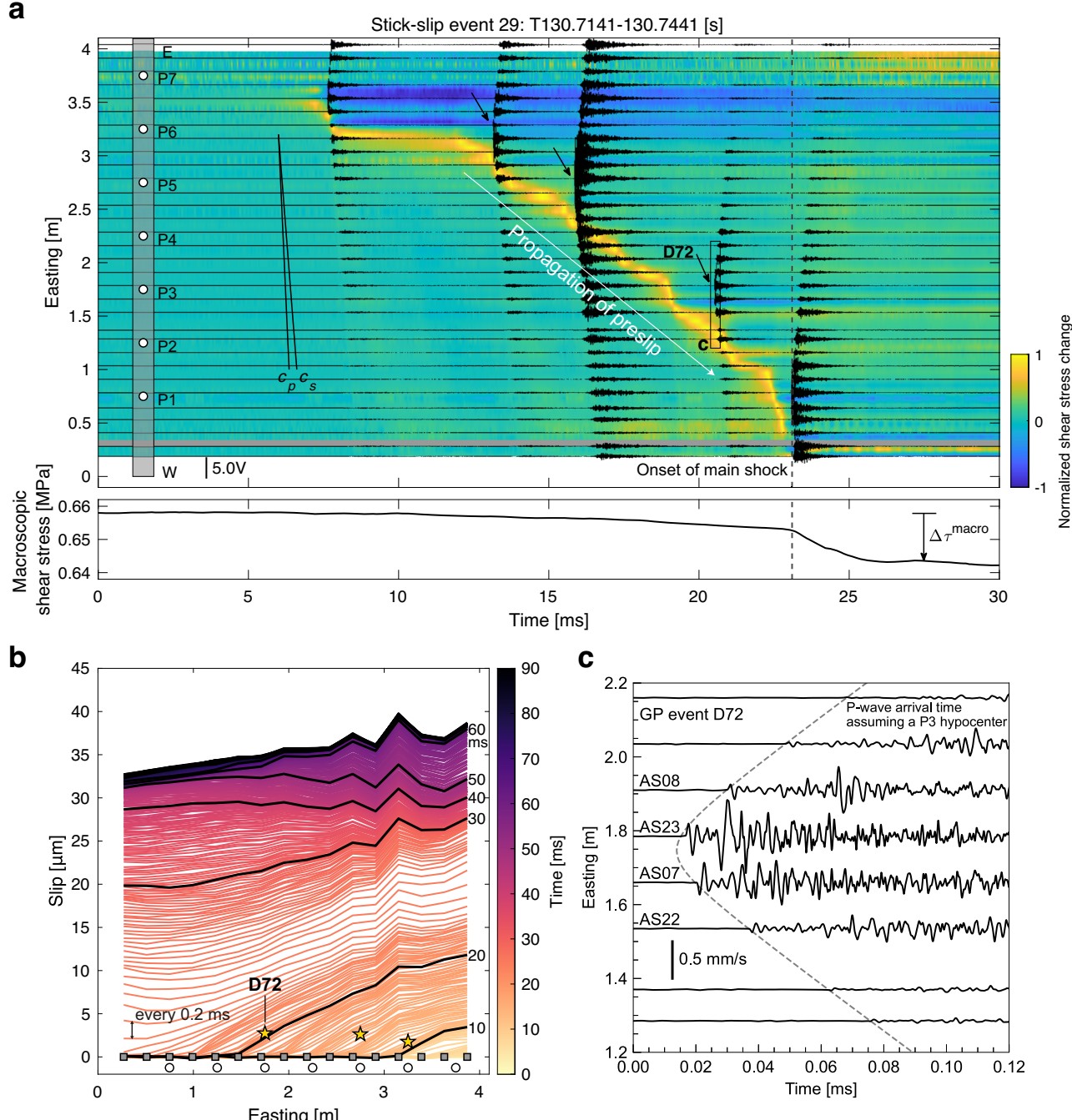

**Fig. 2 | Observation of gouge patch (GP) events during a stick-slip event.**
**a** Evolution of shear stress change and the acoustic emission (AE) waveforms. The color map shows the local shear stress change, and the black lines show the AE waveforms. A strain gauge approximately 0.3 m from the west is missing, indicated in gray. A 10 kHz low-pass filter is applied to the shear stress change, and a 0.1-1 MHz two-way band-pass filter is applied to the AE waveforms to reduce noise. Large main-shock amplitudes are present in the raw data but are clipped in the recording and do not appear after band-pass filtering. The events annotated by the arrow are the GP-generated foreshocks. The lower panel shows the time history of the

macroscopic shear stress measured at the west side of the bottom rock specimen. The vertically dashed line indicates the onset of the main shock, identified by the macroscopic shear stress change. **b** Accumulation of slip during the stick-slip event. Stars indicate the location and timing of the GP foreshocks. The colored and the thick black lines represent the profiles of slip distribution with intervals of 0.2 ms and 10 ms, respectively, with the larger spans corresponding to the coseismic slip of the main shock. **c** AE waveforms associated with GP event D72, as annotated in (**a**). A one-way band-pass filter with a frequency range of 0.1 to 1 MHz is applied. The dashed line indicates the expected P-wave arrival time assuming a hypocenter at P3.

considered in the fitting analysis; however, we averaged the data from the four AE sensors surrounding the GP to minimize its uncertainty. We excluded the farther sensors from the analysis because the signal-to-noise ratio is low, particularly for smaller GP events, which limits a fair comparison across events with different magnitudes.

Figure 3c shows the moment-duration scaling of the GP events generated by P3, derived from the average of four AE sensors and selected based on thresholds related to the fitting quality and signal-to-noise ratio of the P-waveforms (Methods; see Supplementary Table S1 for the observed source parameters). The source parameters

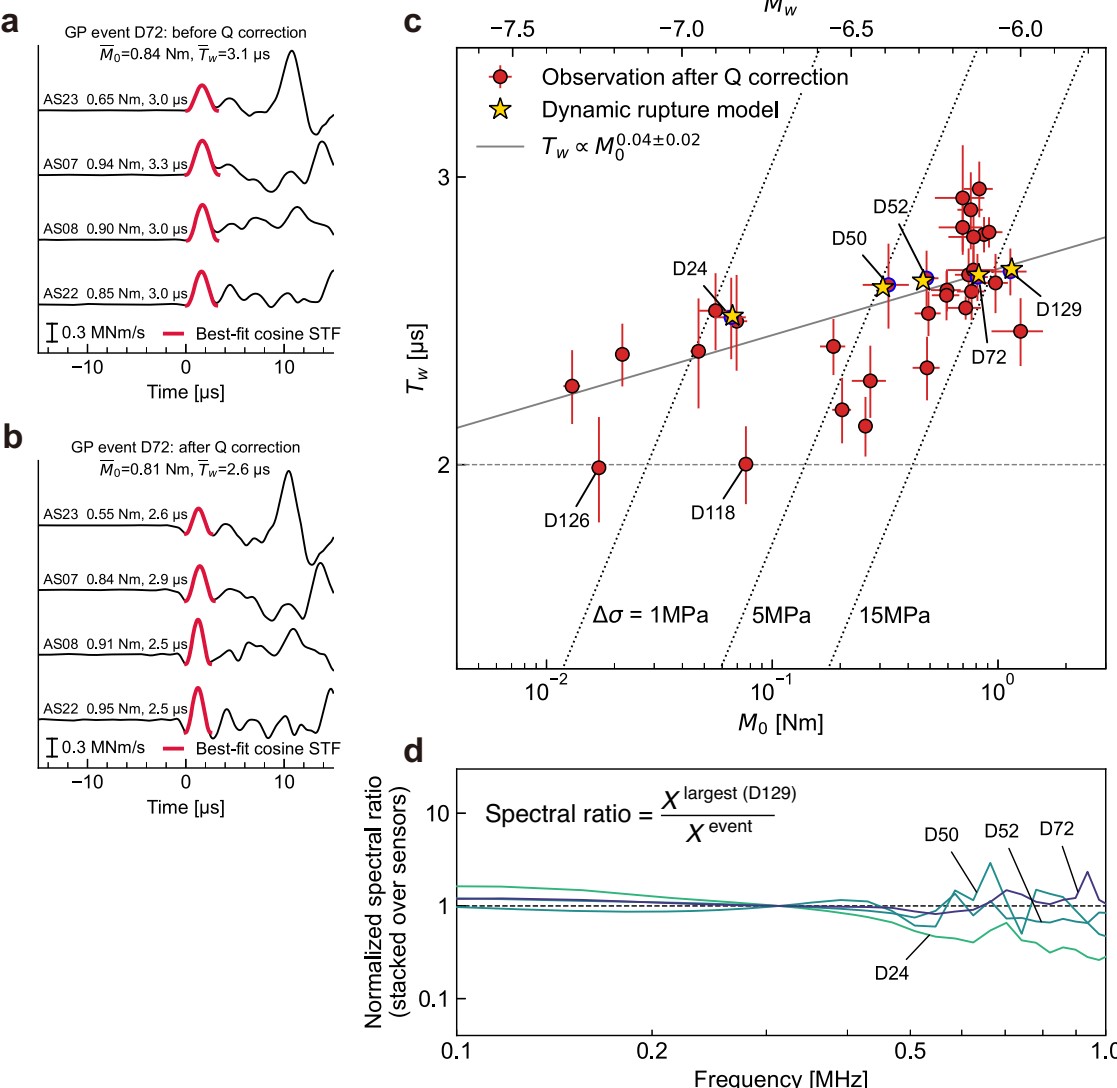

**Fig. 3 | Evaluation of source parameters from source time function (STF) fitting, moment-duration scaling of gouge patch (GP) events, and spectral ratio analysis. a** Observed waveforms (black lines) and the best-fit cosine STFs (thick red lines) shown before attenuation correction. Amplitudes of the observed waveforms were converted from displacement to seismic moment rate using the P-wave radiation pattern and the free-surface reflection coefficient (see "Methods"). Waveforms are aligned by the P-wave arrival times. Source parameters were independently estimated from each of the four acoustic emission (AE) sensors located near the GP and subsequently averaged to obtain representative values. **b** Same as (**a**), but after applying attenuation correction (Methods). The presence of acausal signals likely reflects limitations of the assumed acausal attenuation model.

**c** Moment-duration scaling of GP events after attenuation correction. Circles with error bars represent the mean values and standard errors of the source parameters estimated from the four AE sensors. Stars indicate the modeled target events, while blue-edged circles denote the corresponding observational estimates. Two events (D118 and D126) exhibiting shorter source durations are annotated. The gray solid line shows the best-fit linear regression (major axis method) for the non-self-similar GP event cluster, defined here as all displayed events excluding D118 and D126. Dotted lines indicate the self-similar scaling assuming a constant stress drop and a rupture velocity of $0.9c_s$. **d** Stacked spectral ratio, computed as the mean across the four AE sensors (see Supplementary Note S1 for processing details), and normalized at 0.3 MHz.

of the GP event cluster show a clear deviation from typical self-similar scaling predicted by classical source models. This deviation indicates the non-self-similarity of the GP events. The magnitudes range from $M_w$ −7.3 to − 6.0, with the majority of events exhibiting a source duration $T_w$ of approximately 2.5 μs.

A common objection to non-self-similar scaling is that the observed source duration may be biased by instrumental response or path effects rather than representing source characteristics. One way to rule out this concern is to identify events with shorter durations than those within the non-self-similar cluster[11,15]. The presence of such events indicates that the medium permits the propagation of higher-frequency waves and that the measurement system is capable of resolving them, thereby allowing for reliable estimation of source

parameters. Among the GP events, we identified two such cases, D118 and D126 (Fig. 3c), with source durations of approximately 2.0 μs, which are likely shorter than those of the other GP events. These observations suggest that the non-self-similarity in the GP events reflects the source characteristics rather than artifacts arising from attenuation or instrumental limitations.

The two smaller events (D118 and D126) would reflect limitations inherent to the attenuation-deconvolution process with water-level stabilization. Under these conditions, sources with shorter durations cannot be reliably resolved because attenuation distorts the far-field P-wave displacement pulse beyond the recoverable bandwidth.

For clarity, we define the non-self-similar cluster as all GP events shown in Fig. 3c, excluding the two short-duration events, D118 and

D126, whose durations approach the minimum resolvable limit of the measurement and may therefore bias the scaling analysis. We performed a linear regression on this cluster, yielding a scaling exponent of 0.04 ± 0.02. This value is substantially smaller than the cube-root scaling expected for self-similar earthquakes and is consistent with the non-self-similar scaling reported in natural observations[10,13,14,16]. The variation in the source parameters within the cluster is likely analogous to that observed in the natural repeating earthquakes (e.g.[31–34]). The possible source mechanisms underlying this variability are addressed in the Discussion section.

Another metric to evaluate the detailed scaling of non-self-similar earthquakes is the ratio of their source spectra (e.g.[12,15,16]). We conducted spectral ratio analysis for five representative non-self-similar GP events (D24, D50, D52, D72, and D129). Raw AE waveforms were used without correcting for instrumental response, sensor coupling, or attenuation, as these effects are expected to cancel in the spectral ratio (Supplementary Note S1). The amplitude spectra for the recorded waveforms are shown in Supplementary Fig. S8. The stacked spectral ratios from four AE sensors closest to P3 are shown in Fig. 3d. The nearly flat spectral ratios indicate only minor differences in corner frequency, whereas the decreasing trend observed particularly for D24 is consistent with the weak scaling behavior inferred from Fig. 3c. Although the detailed scaling remains somewhat sensitive to the choice of fitting bandwidth and window functions, the spectral ratio results are compatible with the non-self-similar behavior identified in the time-domain analysis.

We also analyzed the foreshocks and aftershocks generated from the second most activated patch, P5 (Supplementary Fig. S9). The recorded waveforms show high phase coherence, whereas their amplitudes vary across a certain range. While we did not conduct a full source analysis for these events, the observed characteristics indicate that the non-self-similar behavior is not unique to P3 but may be a robust feature of ruptures on the GPs.

## Dynamic rupture modeling of the non-self-similar GP events

To investigate the source mechanisms responsible for the non-self-similar scaling of the GP events, we developed a dynamic rupture model constrained by the estimated source parameters and the controlled GP configuration. Our aim is not to evaluate this model against previously proposed explanations (e.g.[20,22]), nor to argue which mechanism is most representative of natural non-self-similar earthquakes. Instead, we construct a feasible dynamic rupture framework that is consistent with our experimental configuration, quantitatively explains the observed source parameters, and complements previously proposed models. Although some aspects remain incompletely constrained, including the final rupture extent and the shear-traction history on the GP, our objective is to identify the physical factors capable of robustly reproducing non-self-similar scaling in our experimental system and to provide insight into possible mechanisms applicable to natural earthquakes.

The primary challenge in modeling non-self-similar earthquakes is determining how to vary the seismic moment while maintaining a nearly constant source duration. Proposed explanations include varying the stress drop while keeping the source size fixed[10–13,20,21], or accelerating the rupture velocity by slightly adjusting the source size near the critical nucleation size[22].

The final rupture size can, in principle, be constrained through kinematic inversion of static strain or slip changes (e.g.[35]). However, no analyzable static offsets associated with the GP events are detected in either the strain or slip records, likely because the signals fall below the resolution limit of the measurements. While the final rupture extent remains unconstrained by static inversion, we assume that the source patch producing positive stress drop coincides with the GP dimension, and that the final rupture extent is governed by the rupture dynamics under the prescribed barrier conditions.

Under this fixed-source-patch assumption, variations in stress drop are considered as a potential mechanism for the observed differences in seismic moment. We assumed different peak frictions within the patch under the condition that the strength excess and the residual friction level are comparable across all GP events to be modeled, so that the stress drop varies. We then determined the best-fitting peak friction values that reproduce the observed source parameters exhibiting non-self-similar scaling as detailed in the Methods section.

To investigate whether the assumption of variable peak friction is compatible with the observations, we compared the seismic moment of GP foreshocks with the local cumulative slip, defined as the slip surrounding the GP accumulated from the initiation of preslip to the onset of the GP event, inferred from spatiotemporal interpolation of gap-sensor data (Supplementary Fig. S10a). This local cumulative slip can be interpreted as an analogy for slip deficit on the GP (e.g.[36]). The comparison showed that foreshocks with larger cumulative slip tend to exhibit larger seismic moments (Supplementary Fig. S10b). Under our modeling assumption that the seismic moment scales with peak friction, the observed correlation between local cumulative slip and seismic moment does not contradict variability in peak friction among GP events.

We also examined correlations between the observed seismic moment $M_0$ and a set of local and macroscopic measurements, including local slip velocity, macroscopic stress drop, and hold time, to identify factors controlling variations in $M_0$. All evaluated quantities are described in Supplementary Note S2 and summarized in Supplementary Fig. S11.

A possible trend is observed for the total macroscopic accumulated slip, in which smaller events tend to occur at earlier stages of the experiment, while larger events become more frequent at later stages (Supplementary Fig. S11f). This tendency may reflect progressive evolution of the physical state of the gouge patch as macroscopic slip accumulates on the fault. However, none of the other additionally examined measurements show a clear correlation with $M_0$. Consequently, these quantities do not provide additional constraints that can be incorporated into the development of the dynamic rupture framework for the GP events.

Regarding rupture arrest mechanisms, previously proposed variable stress drop models typically assume a strong rupture barrier surrounding the source region, which constrains the final rupture size regardless of coseismic rupture dynamics. While such a setting is plausible for a source patch surrounded by a velocity-strengthening creeping fault, such as the San Andreas Fault[27,37], our experimental setup differs. The area surrounding the GP consists of a smoothed bare rock contact, which is unlikely to act as a strong rupture barrier. To address this discrepancy, we incorporated a relatively weak barrier around the source patch in our source model, making it more consistent with our experimental conditions.

We applied a linear slip-weakening friction law to the GP, instead of a rate-and-state formulation, to simplify the model setup. Under this formulation, the weak barrier allows the rupture to expand beyond the source region, analogous to rupture penetrating a velocity-weakening barrier (e.g.[38]), particularly for events with large stress drops. Since both seismic moment and source duration increase with rupture area, as predicted by classical source models, this behavior is not compatible with the non-self-similar scaling (Supplementary Fig. S12).

In principle, the observed source parameters could be matched by varying the nucleation location within the GP or by prescribing different final rupture extents. We do not exclude these scenarios; however, in the absence of sufficient measurement resolution to constrain the nucleation location from far-field P-wave onsets or to infer the final rupture size from static strain changes, such approaches would require substantial event-by-event tuning of source characteristics. In our view, this would limit the generality of the modeling strategy.

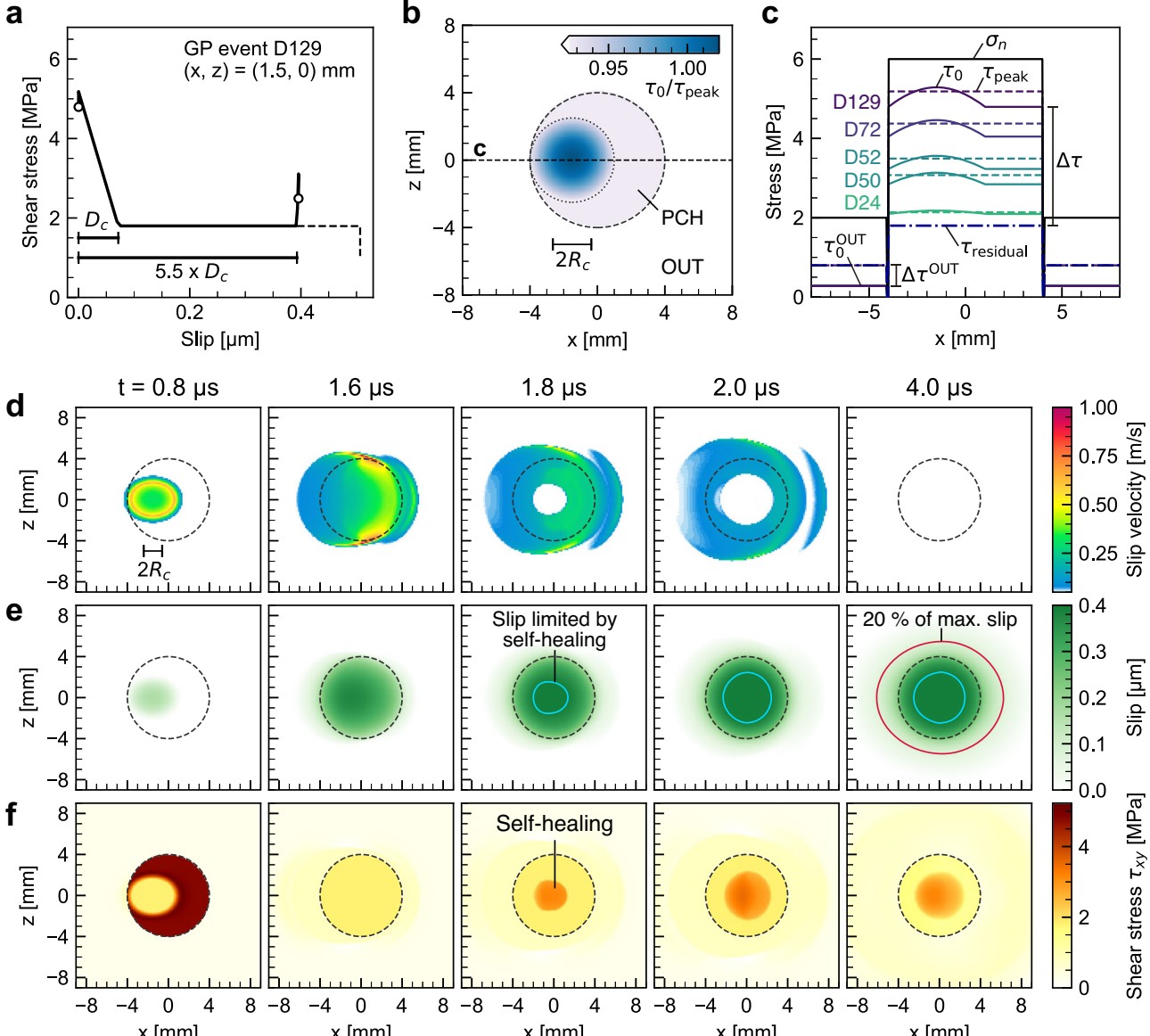

**Fig. 4 | Dynamic rupture models associated with non-self-similar gouge patch (GP) events. a** Linear slip-weakening friction law with self-healing used in the dynamic rupture models. Solid and dashed lines represent the shear traction histories for the cases with and without self-healing friction, respectively. White circles indicate the beginning and end of the shear traction evolution. The slip-strengthening was imposed at 5.5 times $D_c$ ("Methods"). **b** Initial condition of the shear stress distribution. The source patch (PCH) and the surrounding outer region (OUT) are indicated. The color contour shows the initial shear stress normalized by the peak strength. Rupture is nucleated where the initial shear stress exceeds the frictional strength within the nucleation region, shown by the dotted line inside the PCH. The mean critical nucleation size $2R_c$ for the five target events is annotated. **c** Profile of the assumed shear and normal stress distributions associated with the five gouge events along the line at $z = 0$, as indicated by the horizontal dashed line in (**b**). The solid black and colored lines represent the initial normal and shear stresses, respectively, while the dashed and dash-dotted lines indicate the peak and residual friction strengths, respectively. Note that both the peak and residual friction coefficients in the OUT region are identical and set as constants ("Methods"). **d** Snapshots of slip velocity for the target event D129. **e** Same as (**d**) for cumulative slip. The light-blue contour delineates the region where slip is limited by self-healing friction. The final rupture area, evaluated as the 20% slip contour, is shown by the red line. **f** Same as (**d**) for shear stress. Self-healing friction is activated at the center of the patch region.

We therefore incorporated a self-healing friction term as an additional factor in the dynamic rupture model (Fig. 4a). This mechanism suppresses excessive slip at the center of the GP that would otherwise result from rupture expansion into the weak barrier region, thereby limiting the elongation of the source duration. This behavior would be consistent with the analyses of Wang and Day[39] and Kano et al.[40], which demonstrated that pulse-like ruptures exhibit higher corner frequencies than crack-like ruptures under comparable rupture velocities. From a mechanical perspective, self-healing is expected under velocity-dependent friction laws (e.g.[41]) and has been

documented in rock friction experiments (e.g.[42–44]), providing a relevant analog to natural fault conditions. In our implementation, this effect is represented by introducing a linear slip-strengthening part after a prescribed slip distance $D_s$ ("Methods").

We selected five GP events (D24, D50, D52, D72, and D129) showing typical non-self-similar scaling as modeling targets (Fig. 3c). We first conducted multiple simulations to find a feasible range of model parameters, including the strength excess coefficient and barrier efficiency ("Methods"). We subsequently optimized the peak friction on the GP through a grid search to quantitatively reproduce the

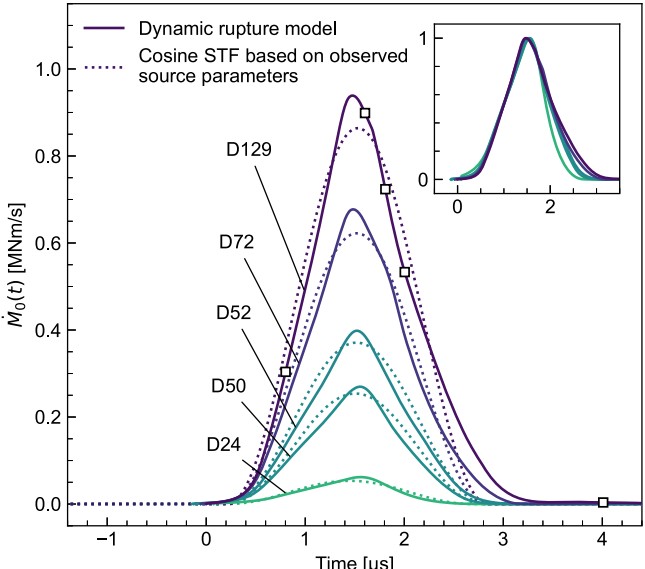

**Fig. 5 | The source time functions (STFs) obtained from the dynamic rupture models for the five target gouge patch (GP) events.** Solid lines show the STFs derived from the dynamic rupture models for the target events, whereas dotted lines indicate cosine STFs constructed from the observed source parameters, averaged across the four acoustic emission (AE) sensors after the attenuation correction. Square markers on the STF of event D129 denote the snapshot timings shown in Fig. 4d–f. The inset displays amplitude-normalized STFs, highlighting the non-self-similarity of the dynamic source process.

observed seismic moments and source durations of the target events. The resulting best-fit parameters are summarized in Supplementary Table S2, and their spatial distributions are shown in Fig. 4b, c. We allowed the characteristic slip distance of the slip-weakening law, $D_c$, to increase with the magnitude of the target events in order to moderate the stress-drop-dependent moment accelerations in the modeled STFs and to achieve quantitative agreement with the observed source parameters. Details of the systematic approach for model parameter selection are provided in the Methods section.

The dynamic rupture process on the GP event associated with D129 is shown in Fig. 4d–f. The dynamic rupture model estimated the final rupture area, defined as the contour of 20% of the maximum slip, to be 1.5 and 1.4 times the patch region in the Mode II and III directions, respectively, assuming a given barrier efficiency.

The STFs obtained from the optimized dynamic rupture models, derived by integrating the slip velocity time history ("Methods"), closely reproduced the cosine STF synthesized using the averaged source parameters estimated from the four AE sensors near the GP (Fig. 5), capturing the observed non-self-similar scaling. The source parameters, corrected for attenuation effects in the observational analysis (Methods), are listed in Supplementary Table S3 and show quantitative agreement with the target event scales (Fig. 3c). The amplitude-normalized STFs (inset of Fig. 5) further emphasize the non-self-similarity of the dynamic source process in the target events.

The key mechanism responsible for the non-self-similar behavior is the suppression of excessive slip at the patch center by self-healing friction (Fig. 4e). In the absence of healing, slip growth localizes at the patch center, particularly under low barrier efficiency. This localization is suppressed once self-healing friction is activated, limiting further slip accumulation. In addition, under the assumption that the prescribed slip distance for the healing part, $D_s$, scales with $D_c$, the extent and magnitude of slip suppression increase with the stress drop within the patch (Supplementary Fig. S13), which in turn leads to tighter STFs for larger events. Deviations from the elliptic slip distribution

predicted by the circular crack model have been reported by Wang and Day[39]. Kano et al.[40] further showed that self-healing slip pulses can generate an additional high-frequency characteristic, reflected in double-corner frequencies, consistent with the physical interpretation of our modeling framework. Taken together, these effects result in nearly constant source durations across the target events.

Overall, these results indicate that a dynamic rupture model incorporating variable stress drop on an isolated patch and self-healing friction provides one feasible, quantitative, and internally consistent explanation for the observed non-self-similar scaling. This framework complements previously proposed mechanisms and thereby broadens the range of conditions under which non-self-similar scaling can emerge.

## Discussion

The GP-generated events showed non-self-similar scaling (Fig. 3c), deviating from the $T_w \sim M_0^{1/3}$ relationship predicted by the classical source models that assume constant stress drop and constant rupture velocity with variable source size. The non-self-similarity persists even after correcting for the amplitude and frequency responses of the sensors and the intrinsic attenuation of the rock specimen, suggesting that the observed scaling likely reflects the true source process rather than measurement artifacts. The precision of these corrections shows an advantage of laboratory experiments, particularly in settings where attenuation effects can be independently constrained.

The proposed model, which assumes a fixed source patch size with variable stress drop, indicates that self-healing friction can effectively shorten the source duration of larger events on a fixed source patch, even in the absence of a strong rupture barrier. As a consequence, this model complements existing frameworks and broadens the range of tectonic settings under which non-self-similar earthquakes may occur. Moreover, the GP events provide insights into the earthquake generation processes not only on tectonic faults but also in volcanic environments (e.g.[45]) and at subglacial asperities on ice-bedrock interfaces[46].

Given the variation of peak friction required in the proposed dynamic rupture model, the potential micromechanisms may originate from differences in the formation of force chains[47–49], or from differences in slip interfaces within the gouge layer or at the boundary between gouge and host rock.

The source mechanisms for the GP events outside the set of modeling targets can be interpreted according to the degree to which their source parameters deviate from those of the modeled events. Events whose source parameters closely match those of the targets could likely be reproduced with modest adjustments to the stress or frictional conditions. In contrast, events that show more substantial deviations may require modifications to the underlying modeling assumptions. Possible factors include partial ruptures confined within the GP due to preceding aseismic slip[50,51]; variations in barrier efficiency outside the GP; differences in nucleation locations within the GP[6,52]; and variations in how $D_c$ and $D_s$ scale with the seismic moment of GP events. Additional experimental constraints will be required to refine the dynamic rupture models for these possibilities.

Further numerical investigation using a rate-and-state friction framework would be worthwhile, as self-healing behavior can arise naturally from the constitutive law (e.g.[41]). Dynamic rupture models that include a velocity-weakening or velocity-neutral region surrounding the source patch, together with variations in stress drop within the patch, may provide additional support for the role of self-healing friction in the emergence of non-self-similar scaling in the absence of a strong rupture barrier.

Experiments using larger GPs will be crucial for further validating the source mechanisms of non-self-similar earthquakes. Although the observed scaling lies above the measurement limit on source duration, imposed by the water-level effect during attenuation deconvolution,

larger patches would produce longer source durations that can be clearly separated from this limitation and would also clarify whether partial ruptures occur within the patch. Larger patches would additionally allow the nucleation location to be constrained by analyzing P-wave travel times, providing further constraints on the dynamic rupture modeling.

Natural observations of non-self-similar earthquakes can provide a valuable perspective for clarifying the overall scaling laws of earthquakes, bridging laboratory and natural fault systems. A conceptual framework is that the characteristic time scales are governed by asperity size, such that fault systems hosting asperities of various dimensions collectively exhibit self-similar scaling, as predicted by classical source models. In contrast, for events associated with a given asperity size, variations in seismic moment can occur while the characteristic timescale remains nearly constant, leading to non-self-similar scaling as investigated in this study. This conceptual scaling law, encompassing both self-similar and non-self-similar behaviors, is illustrated in Supplementary Fig. S14.

The non-self-similar earthquakes may also help identify fault planes based on their source locations and enable monitoring of the evolution of aseismic slip on faults, as has been done in studies of repeating earthquakes. In addition, a more detailed analysis of the scaling exponent within non-self-similar clusters, when evaluated against plausible source models, has the potential to yield further understanding of the mechanical conditions of asperities and their surrounding fault regions.

## Methods

### Setup for local measurements on the fault

We measured the motion normal to the side surface of the rock specimen using AE sensors (V103-RM, Evident Scientific), which have a resonance frequency of 1 MHz. The sensors were glued to the rock surface using cyanoacrylate adhesive over a thin epoxy base, and the signals were amplified by a factor of 100 with a preamplifier (2/4/6C, MISTRAS Group, Inc.). The waveforms were continuously recorded at 10 MHz sampling with 16-bit resolution. Slip measurements were obtained using eddy current gap sensors (FK-202F, Shinkawa Electric Co., Ltd.), recorded at 50 kHz sampling with 24-bit resolution. To monitor local strain variations near the fault, we used biaxial and triaxial semiconductor strain gauges (KSN-2-120-F3-11 and SKS-30282, Kyowa Electronic Instruments Co., Ltd.). The strain signals were processed using a signal conditioner (CDA-700A/CDA-900A, Kyowa Electronic Instruments Co., Ltd.) and recorded at 1 MHz sampling with 16-bit resolution. The strain offset was initialized when the top rock specimen was lifted by the center hole jacks. For local measurements, we used 32 AE sensors, 32 biaxial and 32 triaxial strain gauges, and 16 gap sensors (Supplementary Fig. S2). Data acquisition for these measurements began simultaneously and was kept synchronized by referencing a common 10 MHz clock pulse. Further details regarding the friction apparatus can be found in Yamashita et al.[26].

### Setup for fault surface state and gouge patches

The contact surfaces of the rock specimens were initially polished such that the undulation of the surface was less than 30 μm[26]. In addition, they have been further lubricated after tens of stick-slip experiments with various normal stresses ranging from 2 to 6 MPa for nearly 80 mm of slip in total. Eventually, foreshocks caused by the natural heterogeneity of the fault were not generated as frequently as Yamashita et al.[26], who used the same apparatus as this study and reported the generation of foreshocks during the evolution of preslip. This smoothed fault surface was preferable for identifying GP events.

The GPs were located on the fault using a sieve of 100 μm and a circle ruler. We smoothed its topography by pressing it with a plastic sheet to moderate normal stress concentration. Under a macroscopic uniaxial load of 2 MPa, the average and local maximum normal stresses were approximately 4.5 MPa and 6.5 MPa, respectively, estimated using a pressure-sensitive film (Prescale LW, Fujifilm) with a centimeter-scale rock specimen (Supplementary Note S3.1 and Supplementary Fig. S15a, b). The average height within the GP was ~20 μm (Supplementary Note S3.2 and Supplementary Fig. S15c). The median particle diameter of the gouge used in the experiment was 8.2 μm (Supplementary Fig. S15d).

### Creating catalog of GP events

We manually picked the GP events with a high signal-to-noise ratio sufficient to determine the first P-wave arrival time and polarity using at least three AE sensors. Events that were clipped due to signal over-range were excluded. We then relocated the GP events using manually picked P-wave arrival times, by grid-searching for the minimum variance between the observed and theoretical arrival times at four nearby AE sensors, assuming a uniform velocity structure (Supplementary Table S4), following Yamashita et al.[53]. The average relocation error, estimated from the standard deviation of the arrival time residuals multiplied by the P-wave velocity, was approximately 2.4 mm.

### Calibration of AE sensor response

We estimated the transfer function (TF) of the AE sensor, represented by the poles and zeros in the auto-regressive with external input (ARX) model, to correct for its amplitude and frequency response[29,54]. We recorded waveforms, generated by a piezoelectric transducer (10 mm diameter, 1 MHz resonant frequency), excited by a step input, propagating through a steel block of 100 mm x 100 mm x 127 mm (Supplementary Fig. S16). Measurements were taken alternately at the same location using a laser Doppler vibrometer (LDV, Melectro V100, Denshi Giken Co., Ltd.), which is assumed to have a flat response, and an AE sensor. The recorded data were used to constrain the TF parameters via the least-squares method (Supplementary Note S4). The waveforms were stacked 10,000 times for the LDV and 2000 times for the AE sensor to increase the signal-to-noise ratio (Supplementary Fig. S17). The Akaike Information Criterion (AIC) was applied to determine the optimal number of poles and zeros, given by:

$$\text{AIC} = N \ln \frac{1}{N} \sum_{k=1}^{N} (y_k - \widehat{y}_k)^2 + 2(m + n + 1), \qquad (1)$$

where $N$ is the number of discrete time steps, $y_k$ is the system response (the waveform recorded by the AE sensor at time step $k$), and $\widehat{y}_k$ is the one-step predictor obtained using the modeled TF and the time history of waveforms recorded by both the AE sensor and LDV (Supplementary Note S4). The parameters $m$ and $n$ denote the number of poles and zeros, respectively, which were determined to be 24 and 11 (Supplementary Fig. S18). Note that the AE sensor was calibrated to simulate a response to the velocity of surface motion of the rock specimen.

Supplementary Fig. S19a shows the Bode plot of the TF. The amplitude and frequency response of the AE sensor, obtained by comparing the output spectrum of the AE sensor with the input vibration signal inferred from the LDV measurements, are well reproduced by the TF with the optimized poles and zeros. To correct the waveforms of the GP events, we computed the frequency response of the TF using `scipy.signal.freqz`, and performed deconvolution following a process similar to `remove_response` implemented in `obspy`[55]. This correction was validated using additional waveform pairs recorded at different locations on the block. The AE waveforms after the correction matched those from the LDV, suggesting the robustness of the sensor calibration with the optimized TF (Supplementary Fig. S19b). Note that the AE sensor has a sensitivity of ~40 V/(m/s).

## Sensor coupling effect

To evaluate the in-situ sensor coupling factors that compensate for the amplitude variations due to the contact conditions of the AE sensors, we used waveforms generated by steel ball-drop impacts. This method is widely used for sensor calibration[56,57]. A steel ball with a 2 mm diameter was dropped from a height of 0.5 m onto the center of the fault at 32 locations, spaced 120 mm apart (Supplementary Fig. S20). Each drop was repeated three times at each location to confirm the reproducibility of the waveforms. Slight deviations from the intended impact positions were corrected by relocating the sources based on P-wave arrival times, using an auto-picking function[58].

To estimate the true motion of the rock surface caused by the ball-drop impacts, we simulated waveform propagation using the finite-difference-based software `OpenSWPC`[59] (Supplementary Note S5). The software was extended to detect the free surfaces along the boundaries of the rock specimen. The extended version was cross-verified by comparing it with the wavenumber integration method in `Computer Programs in Seismology`[60], which calculates Green's functions for an infinite plate model (Supplementary Figs. S21, S22). The force-time function of the ball-drop impact was estimated using Hertzian impact theory[61,62].

The sensor coupling factors were optimized by minimizing the residuals between the observed and simulated P-wave amplitudes, following the method of Kwiatek et al.[30] (Supplementary Note S6 and Supplementary Fig. S23). Each AE sensor was calibrated using data from at least three nearby ball-drop sources, except for the sensors located at the eastern and western ends of the specimen. After applying geometric constraints between source and sensor positions, we used 117 source-sensor pairs in the optimization.

We also applied a correction for the aperture effect, which becomes significant when the incident wavelength is comparable to or smaller than the sensor contact diameter[29,57,63]. The amplitude correction factor is given by:

$$\beta(\omega, \theta) = \frac{2v_a}{\omega R} J_1\left(\frac{\omega R}{v_a}\right), \tag{2}$$

where $\omega$ is the angular frequency, $\theta$ is the incident angle, $v_a = v/\sin\theta$ is the apparent wave velocity, $J_1(x)$ is the first-order Bessel function of the first kind, and $R$ is the sensor radius (6.35 mm in this study; Supplementary Fig. S24). Further details are provided in Supplementary Note S7. The angular frequency $\omega$ was also optimized together with the coupling factors.

To address the trade-off between the sensor coupling factors and the intrinsic attenuation of the rock specimen, we first determined the coupling factors, assuming a constant quality factor, $Q_p^{const} = 200$, in the simulated waveforms. Using these coupling factors, we then refined the attenuation model to incorporate frequency-dependent attenuation, $Q_p^{-1}(f)$, as described in the following section.

The sensor coupling factors were determined to be $0.81 \pm 0.20$ (Supplementary Fig. S25), indicating that the observed P-waveform amplitude in velocity was reduced by approximately 20% compared to the numerically modeled waveforms. Applying this correction improves the accuracy of seismic moment estimates.

## Attenuation

We investigated the frequency-dependent attenuation factor by comparing the observed and simulated P-wave spectra generated from the ball-drop impact, using the same dataset as for sensor coupling calibration. The attenuation can be written in terms of the quality factor $Q_p^{-1}(f)$ as follows (e.g.[64]):

$$\widetilde{y}_{obs}(f) = \widetilde{y}_{model}(f) \exp\left(-\frac{\pi f r}{Q_p(f) v_p}\right), \tag{3}$$

where $\widetilde{y}_{obs}(f)$ is the observed P-wave spectrum after correction for the sensor response, coupling factor, and aperture effect (Eq. (2)). $\widetilde{y}_{model}(f)$ is the spectrum of the numerically modeled waveforms without attenuation. The parameter $r$ denotes the source distance, and $v_p$ is the P-wave velocity of the rock specimen.

For preprocessing, we first demeaned and detrended the observed and modeled P-wave pulses, time-shifted them to maximize their correlation, and then applied tapering using a Hann window (Supplementary Fig. S7a). We then computed the P-wave spectra and convolved a Hann window with a length of 7 data points to further smooth the spectra, following the spectral smoothing technique[65]. The spectral ratio $\ln|\widetilde{y}_{obs}(f)/\widetilde{y}_{model}(f)|$ provides the attenuation estimates associated with each source-sensor pair (Supplementary Fig. S7b).

The quality of the P-wave spectral comparison for each source-sensor pair was evaluated using variance reduction (VR) of the P-waveforms in the time domain, defined as:

$$VR = 1 - \frac{\sum_{k=1}^{N}(y_{obs}[k] - y_{model}[k])^2}{\sum_{k=1}^{N} y_{obs}^2[k]}. \tag{4}$$

We selected 67 out of 117 source-sensor pairs for analysis, applying a VR threshold of 0.95.

We calculated the statistics of $Q_p^{-1}(f)$ across frequencies for the selected source-sensor pairs and derived the median attenuation model shown in Supplementary Fig. S7c. This model provides an averaged estimate of attenuation for the rock specimen, helping to reduce the bias caused by the path effects in the source duration estimates of the GP events. We deconvolved the median attenuation model from the P-waveforms, applying a water level set at 30% of the maximum spectral amplitude.

## Fitting STF

The P-displacement waveforms of the GP events were obtained after applying corrections for the sensor response, coupling factors, and attenuation effects. To mitigate high-frequency artifacts introduced by deconvolution, a two-way low-pass filter with a cutoff at 1 MHz was applied before the attenuation correction. These corrected velocity waveforms were then integrated to obtain the displacement component normal to the side surface of the rock specimen. The amplitude of the resulting displacement was subsequently converted to the seismic moment rate following the approach outlined by Yamashita et al.[26], incorporating the radiation pattern and the reflection coefficient at the free surface ([64], Q5.6). Finally, a synthetic cosine-type STF (e.g.[66]) was fitted to estimate the source parameters.

The synthetic STF was characterized by three parameters: seismic moment, source duration, and a time shift ($T_{shift}$) to align the onset of the P-wave pulse. These parameters were optimized for each GP event-sensor pair by minimizing the root-mean-square error (RMSE) between the observed and synthetic STFs using `scipy.optimize.minimize` with the Nelder-Mead method. The fitting window was restricted to 75% from the STF onset to minimize bias from later parts of the P-wave pulse, which may be affected by scattering. Parameter ranges were set as (0, 10) Nm for $M_0$, (1, 10) μs for $T_w$, and (−0.5, 1) μs for $T_{shift}$. The quality of source parameters was evaluated using two criteria: (1) the RMSE of the fitting, normalized with the maximum amplitude of the best-fit STF, was required to be less than 0.06, and (2) the signal-to-noise ratio of the observed P-wave, defined as the ratio of the maximum P-wave amplitude to the standard deviation of the noise measured in the 10 μs preceding the P-wave onset, had to be greater than 4.5. We included GP events in the statistical analysis of scaling only if all four sensors met these thresholds, resulting in the selection of 33 out of 44 GP events generated by P3 (Supplementary Fig. S26).

## Dynamic rupture modeling

The geometrical configuration of the GP on the fault was modeled as a circular source patch (PCH) with a diameter of 8 mm, including a nucleation zone, and a separate outer region (OUT) surrounding the patch (Fig. 4b). The normal stress ($\sigma_n$) was set to 6 MPa in the PCH and 2 MPa in the OUT, based on estimates from pressure-sensitive film measurements (Supplementary Fig. S15b). A 0.08 mm gap, possibly due to a topographic step between the PCH and OUT, was modeled as a stress-free region.

A linear-slip weakening law was applied to the PCH, where the peak friction coefficient ($\mu_s$) and the characteristic slip distance ($D_c$) varied depending on the target event magnitude. The residual friction coefficient ($\mu_d$) was fixed at 0.3 for the sake of simplicity. The peak and residual frictional strengths were defined as $\tau_{\text{peak}} = \mu_s \sigma_n$ and $\tau_{\text{residual}} = \mu_d \sigma_n$, respectively. In the OUT region, the friction coefficient ($\mu^{OUT}$) was assumed constant at 0.4, representing a residual level under the assumption that the friction was weakened due to the surrounding preslip prior to the onset of the GP events. This value was inferred from the friction coefficient of approximately 0.4 estimated from the macroscopic shear-to-normal stress ratio on the bare rock contact (lower panel of Fig. 2a).

The peak friction coefficient $\mu_s$ was optimized to determine the coseismic stress drop ($\Delta\tau$) within the PCH, expressed as a ratio to the reference stress drop ($\Delta\sigma_{\text{ref}}$) derived from the static crack model. The $\Delta\sigma_{\text{ref}}$ is written as:

$$\Delta\sigma_{\text{ref}}^{(i)} = \frac{7}{16}\frac{M_0^{(i)}}{a_{\text{PCH}}^3}, \tag{5}$$

where the superscript $i$ denotes the model parameters for the $i$th target event, $M_0^{(i)}$ is the seismic moment estimated from the observations, and $a_{\text{PCH}}$ is the radius of the PCH, including the stress-free gap (4.08 mm). The coseismic stress drop was then defined as:

$$\Delta\tau^{(i)} = s^{(i)}\Delta\sigma_{\text{ref}}^{(i)}, \tag{6}$$

where $s^{(i)}$ is a scaling factor that controls the stress drop in the dynamic rupture model and was optimized to fit the source parameters derived from the STF of the dynamic rupture model to those of the observed STF. The initial shear stress was defined as:

$$\tau_0^{(i)} = c\mu_s^{(i)}\sigma_n^{\text{PCH}}, \tag{7}$$

where $c$ is the strength excess coefficient that quantifies the closeness of the initial shear stress to the peak strength (i.e., the difference between peak and initial shear stress). This coefficient was uniformly set to 0.925 within the PCH, in reference to previous numerical models of repeating earthquakes (e.g.[51]). The peak friction coefficient $\mu_s^{(i)}$ was assumed to be spatially uniform within the PCH. While the absolute strength excess varied among events, the fraction $c$ was kept constant (Fig. 4c). As an exception, for the smallest event (D24), $c$ was increased to 0.98 to facilitate rupture nucleation under a relatively small $\Delta\tau$. The stress drop in the PCH was thus given by:

$$\Delta\tau^{(i)} = (c\mu_s^{(i)} - \mu_d)\sigma_n^{PCH}, \tag{8}$$

allowing us to solve for $\mu_s^{(i)}$ as:

$$\mu_s^{(i)} = \frac{1}{c}\left[\frac{s^{(i)}\Delta\sigma_{\text{ref}}^{(i)}}{\sigma_n^{PCH}} + \mu_d\right]. \tag{9}$$

Another key model parameter is a scaling exponent, $\alpha$, which governs the relationship between the characteristic slip distance ($D_c$) and the mean coseismic slip ($\bar{u}_0$) estimated from the static crack model. This scaling is used to mimic the fracture energy scaling with the amount of slip (e.g.[67]). The relationship is expressed as:

$$D_c^{(i)} = \frac{D_c^{\min}}{(\bar{u}_0^{\min})^\alpha}\bar{u}_0^{(i)\alpha}, \tag{10}$$

where $\bar{u}_0^{\min}$ is the mean coseismic slip of the smallest target event (D24), and $D_c^{\min}$ is the optimized $D_c$ corresponding to this event. The mean slip $\bar{u}_0^{(i)}$ is given by:

$$\bar{u}_0^{(i)} = \frac{M_0^{(i)}}{GA_{\text{PCH}}}, \tag{11}$$

where $A_{PCH} = \pi a_{PCH}^2$ and $G$ is the shear modulus of the host rock specimen (see Supplementary Table S4 for the elastic constants). We determined $D_c^{\min}$ by referring to the fracture energy scaling compiled by Cocco et al.[67] and set $\alpha = 0.6$ based on multiple simulations to identify a plausible parameter set that fits the STF.

The rupture was initiated within a 2.5 mm nucleation radius in the PCH, where $\tau_0$ increases following a Gaussian distribution (Fig. 4b). The peak value of $\tau_0$ was set to $1.02\mu_s^{(i)}\sigma_n$. The perturbation of $\tau_0^{(i)}$ within the nucleation region was neglected when evaluating $\mu_s^{(i)}$ from Equations (8) and (9). The critical nucleation radius ($R_c$) in the PCH was determined using the power-law relationship with the seismic ratio[68], yielding $R_c$ values ranging from 0.9 to 1.6 mm (Supplementary Table S2).

To configure the initial stress condition of the OUT region, we assumed that preslip in the OUT region aseismically releases the accumulated shear stress from interseismic loading prior to the onset of GP events. This is expected to lead to a low initial shear stress level, which acts as a barrier to arrest the rupture by a negative stress drop[69]. The contrast in stress drop between the PCH and OUT is a key factor in determining barrier efficiency (e.g.[6,22,38]). We estimated the initial shear stress in the OUT region as a fraction of its friction strength, such that $\tau_0^{OUT} = \gamma\mu^{OUT}\sigma_n^{OUT}$, where $\gamma < 1$. Accordingly, the stress drop in OUT is given by:

$$\Delta\tau^{OUT} = (\gamma - 1)\mu^{OUT}\sigma_n^{OUT}. \tag{12}$$

To incorporate a relatively weak barrier, which is considered plausible for smoothed bare rock contact, we set $\gamma$ to 0.35, which resulted in rupture penetration beyond the PCH (Fig. 4e).

We modified the linear slip-weakening law to incorporate self-healing friction as follows (e.g.[70]):

$$\mu(\delta) = \begin{cases} -\frac{(\mu_s - \mu_d)}{D_c}\delta + \mu_s, & 0 \leq \delta < D_c \\ \mu_d, & D_c \leq \delta < D_s \\ \frac{k_s(\mu_s - \mu_d)}{D_c}(\delta - D_s) + \mu_d, & D_s \leq \delta \end{cases} \tag{13}$$

where $\delta$ is the coseismic slip, and $D_s$ is the predefined slip distance at which self-healing friction is activated. The stiffness associated with the slip-strengthening rate $k_s$ is set to 10. We assumed $D_s = 5.5D_c$ as a possible value, determined through multiple simulations.

We performed dynamic rupture simulations using the spectral boundary integral method-based software UGUCA[71] (See Supplementary Note S8 for the detailed configurations). The STF of the dynamic rupture model was computed as:

$$\dot{M}_0(t) = G\int_A \dot{\delta}(\boldsymbol{\xi}, t)dA, \tag{14}$$

where the integration area $A$ corresponds to the final rupture extent. For a fair comparison with the observed source parameters, we evaluated $M_0$ and $T_w$ from the dynamic rupture modeling using the same approach as that used in observational analyses. Specifically, we first

convolved the modeled STF with an attenuation factor to mimic the path effects. We then deconvolved the STF using a water-level approach before fitting the synthetic STF to the observed data for source parameter estimation.

To determine the optimal initial stress state that best reproduces the source parameters of the target events, we performed a grid search over $s^{(i)}$. The optimal fit was achieved by minimizing the residual function:

$$\min\left[\frac{(M_0^{obs} - M_0^{model})^2}{\sigma_{M_0}^{model\,2}} + \frac{(T_w^{obs} - T_w^{model})^2}{\sigma_{T_w}^{model\,2}}\right], \qquad (15)$$

and $T_w^{model}$ obtained from the grid search. These normalizations ensure that the residuals for both source parameters are weighted equally. The value of $s^{(i)}$ was constrained to a range of 0.4-0.7 for the target events (Supplementary Table S2).

## Data availability
The datasets containing the AE waveforms, local strain and slip, and other macroscopic measurements are available in the Zenodo repository at https://doi.org/10.5281/zenodo.15233278. The processing codes and Jupyter notebooks for post-processing are documented in the GitHub repository `4mNonSelfSim_Paper` at 10.5281/zenodo.15288780.

## Code availability
For AE sensor calibration, we developed the software tool `AEsensor_Calibration_ARX` to estimate the ARX model, available at https://doi.org/10.5281/zenodo.8359992. The numerical simulations of waveform propagation were performed using an extended version of `OpenSWPC` v5.1.0[59], modified to implement free-surface conditions along the boundaries of the rock specimen. The code is maintained in the GitHub repository `4mNonSelfSim_OpenSWPC` and archived at 10.5281/zenodo.15288446. Dynamic rupture simulations were conducted using `UGUCA` v1.0[71] with self-healing friction. The additional modules and input files for these simulations are available in the GitHub repository `4mNonSelfSim_UGUCA`, archived at 10.5281/zenodo.15288456. AE waveform data were processed using the Python toolbox `ObsPy`[55].

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

## Acknowledgements

This work was supported by the NIED research project "Research and Development for Comprehensive Understanding of Earthquake Generation and Forecasting" (K.O., F.Y., and E.F.) and by JSPS KAKENHI Grant Numbers JP23K22592 (E.F., K.O., and F.Y.), and JP21H05200 (F.Y. and K.O.).

## Author contributions

K.O. designed and performed the experiments, analyzed the experimental data, developed the software, performed the dynamic rupture modeling, and wrote the original manuscript. F.Y. developed the measurement systems, installed the sensors, contributed to the experimental design and implementation, and provided advice on the modeling. E.F. developed the large-scale biaxial rock friction apparatus used in this study and supervised the project. All authors contributed to the revision of the manuscript.

## Competing interests

The authors declare no competing interests.
