## [Transparent Peer Review file · Nature Communications]

Dynamics of non-self-similar earthquakes illuminated by a controlled fault asperity

Corresponding Author: Dr Kurama Okubo

Version 0:

Reviewer comments:

Reviewer #1

(Remarks to the Author)

Review of 'Dynamics of non-self-similar earthquakes illuminated by a controlled fault asperity' by Kurama Okubo, Futoshi Yamashita and Eiichi Fukuyama.

This article investigates the scaling of the source parameters of generated earthquakes during a laboratory experiment analog to a fault frictional slipping process. The authors build upon an already existing setup in order to record generated acoustic signal originating from some designed irregularities over the fault plane that have distinct frictional property. They identified from the recorded acoustic signals that the variation of moment of the recorded ruptures span a large range of moment while at the same time the rupture duration is almost constant. The authors propose that these results contradict the scaling predicted from the classical crack model where the rupture duration should vary with the moment of the recorded rupture. They propose a dynamic rupture model in order to reproduce the observed results and to suggest a possible physical origin for the observed non self-similar scaling.

I'm very impressed by the quality and precision with which the experiments have been carried out. The experimental set-up is clearly exceptional, with dense, cutting-edge instrumentation that makes it possible to record ruptures in a very convincing way. The care taken with the corrections to the measurements (sensor calibration, coupling coefficient, attenuation measurements) is clearly remarkable and lends real credence to the results presented in this work. The problem studied here is highly relevant and represents a new piece of information on the question regarding the existence or not of non-self similar seismic ruptures. As the authors stated, numerous recent measurements carried out in natural context suggest that these unusual ruptures do exist, but there is still some doubt as to the reliability of these findings due to the difficulty of fully understanding the role of attenuation in these results. This experiment therefore provides an unequivocal answer to the question of whether non-self-similar ruptures are possible. The article is well-written, and the experimental observations are well presented such that the results seems very compelling. However, I'm a little less convinced by the numerical modelling part because it seems to me that a lot of parameters and hypotheses are imposed in a rather ad hoc way, although I have to admit I have less expertise on this part.

Finally, I just think it would have been great if the authors use patches with different sizes. This would have made it possible to validate that on the whole the cubic self-similar model is well represented, but also to validate that non-self-similar ruptures appear on different scales. I understand that this would have required a non-negligible piece of work. Therefore I would be looking forward for these new experiments.

General comments :

Dynamic rupture model – As I understand here the authors propose that variation of the peak friction on the same fault patch can occur between successive ruptures of the same fault patch. The change of this peak friction which controls the magnitude of the rupture therefore provides an explanation for the variable moment observed in the experiments. In my opinion this is a bit the weakest part of this manuscript as actually the authors have no data that corroborate this hypothesis related to the change of the peak friction. The Fig S8 b is not totally convincing as a large scatter is visible. While this model is possible and the results of the model are well analyzed, I'm still a little unsure if this model actually captures what happens on the asperity and wonder if other explanations are possible to explain this lack of self-similarity.

To continue on my previous comment, It would have been very interesting, given the amount and the quality of data acquired

during these experiments, to link the variation of the observed stress drop with the recorded parameters; local slip speed, normal stress, shear stress, total cumulative slip (not only the preslip), etc.. This would have been very informative and could have revealed what measurable physical parameter controls this variation of the stress drop.

Could you show the spectra of at least one of the events with a low rupture duration to compare with an event of same moment but with larger rupture duration (for example D118 and M24). I think this should be quite informative for the reader to show that you can highlight the difference between these two spectra.

Most of the analysis and conclusion are drawn from ruptures on patch P3. What about ruptures on the other patches. Do they also show the same behavior as the one reported here?

Below are some more specific comments

L35 : 'classical source model', you mean 'shear crack model'

In the introduction you could also mention the article of Tsai and Hirth, 2019, which propose that the non self-similar ruptures can be explained with a completely different model than the shear crack model (particle impacts). As you know fault gouge particle size you could also see if your results fit this model prediction. As this particle impact model is maybe not the most standard model proposed for the source scaling, I let the authors decide whether or not this analysis is worthwhile

L103 : stick slip event : you mean system size stick slip event? Or could it be just local ones?

L161 four closest sensors ? did you try using the eight closest sensors? Why this choice. Is the signal becoming rapidly to weak or attenuation becomes significant?

L251-252 – does this mean this mechanism is specific to fault gouge experiment?

L263 – 'Dc vary with magnitude of the event', do you mean the size of the patch?, it is not clear to me why Dc should scale with magnitude (for constant size events).

L293-294, Maybe I would avoid this comment that is too general. For example, If you have borehole measurement close to a fault you will be in very similar condition as in laboratory experiment (or maybe even better), regarding the attenuation effect. I think a real strength of these laboratory experiments is that you can control and record many parameters during the process (strain, slip, etc...).

L307-314 I am not sure to understand very well what you really mean here. It seems that you try to support the variation of peak friction between successive stick-slip from a physical basis but it remains quite speculative.

(Remarks on code availability)

Reviewer #2

(Remarks to the Author)

This paper conducts laboratory experiments to investigate non-self-similar earthquakes, where an earthquake's source duration remains nearly constant regardless of its seismic moment, diverging from typical earthquake scaling laws. The authors used a meter-scale laboratory fault with controlled gouge patches to generate microearthquakes exhibiting this behavior. Through careful sensor calibration and attenuation correction, they developed a dynamic rupture model that attributes this non-self-similarity to a combination of variable stress drop and self-healing friction on isolated asperities, suggesting it can occur in diverse tectonic settings beyond previously assumed strong rupture barriers. The study highlights the advantages of laboratory experiments for precise measurements in understanding earthquake mechanisms, addressing a critical question in earthquake physics and allowing for a more accurate estimation of scaling laws and source mechanisms in earthquakes. Here are several suggestions that could further enhance the clarity of the paper.

1. The paper provides a clear definition of non-self-similar earthquakes as "clusters of events that show similar waveforms in phase but vary in amplitude". They then state that while these "may be categorized as repeating earthquakes, the latter term also encompasses events with nearly identical phase and amplitude". However, throughout the Discussion, these terms are often used in close conjunction or somewhat interchangeably (e.g., "variation in the source parameters within the cluster is likely analogous to that observed in the natural repeating earthquakes"; "insights into the generation processes of the repeating earthquakes"; "as has been done in studies of the repeating earthquakes"). Please explicitly state if the non-self-similar events generated in your experiments are considered a subset of repeating earthquakes or if "repeating earthquakes" is primarily used as a broader, less specific term for events with high waveform similarity.

2. The authors use a slip-weakening friction law with self-healing to conduct dynamic rupture modeling. The simulated events maintain constant source duration. Based on the simulation, they claim that variable stress drop, achieved by setting different peak frictions within the gouge patch, is crucial for varying seismic moments, and also reveal that the self-healing friction is "key mechanism for reproducing non-self-similar scaling" by preventing excessive slip. First, why is the slip-weakening plus self-healing law used? Will the more comprehensive rate-state friction law give the same or similar results in

terms of constant source duration and non-self-similar scaling? Second, the simulations set the varying peak friction and self-healing as the input conditions and generate the non-self-healing phenomenon. Logically, this only means that the varying peak friction and self-healing are the causes of non-self-healing, but may not necessarily prove that the varying peak friction and self-healing are the fundamental mechanisms of non-self-healing. This seems a weak reasoning.

3. The authors have summarized previous proposals for non-self-similar scaling, including "fixed source dimensions with variable stress drop or accelerating rupture velocity" and models "based on the isolated asperity patches". The authors' model clearly adopts the "former approach, modeling variations in stress drop". While they briefly contrast their model's lack of a "substantial rupture barrier" with some previous assumptions. Why does their combination of variable stress drop and self-healing friction provide a more robust or broadly applicable explanation for non-self-similar scaling, especially compared to other proposed mechanisms (e.g., rupture acceleration)?

4. The manuscript describes a large-scale biaxial rock friction experiment in which the normal stress is spatially varied—set to 2.0 MPa at the center and reduced to 0.7 MPa at both edges of the simulated fault (Fig. 1b). The authors state that this reduction at the edges is to "promote the smooth nucleation of stick-slip events." However, the rationale for specifically choosing 0.7 MPa as the edge stress value is unclear. Is this value based on prior empirical optimization, numerical modeling, or physical constraints of the setup? Furthermore, what exactly is meant by "smooth nucleation" in this context? How do you define "smooth nucleation" and how will this influence the results in the present paper?

5. Page 4, line 90, "We selected the size of the GP as a diameter of 8 mm, which was preferred as the fraction of the size to the simulated fault was small enough to mimic the source embedded in an infinite medium at depth, compatible with the natural condition." The manuscript states that an asperity diameter of 8 mm was selected to mimic a buried seismic source in an infinite medium. However, the basis for this specific choice is unclear. Was it determined by empirical evidence, prior experiments, scaling analysis, or simply practical considerations? The term "preferred" is vague and lacks scientific justification.

6. The slip-weakening friction law shown in Fig. 4c has a very flat stable friction and a very sharp increase in self-healing. I usually see people using a smooth transition of friction in such a law, i.e., a smooth increase in healing. Will the sharp change in healing influence the results?

7. The magnitudes of the events generated are within the range of M_w -7.3 and -6.0. Will such a narrow magnitude range bias the obtained scaling?

8. The source duration estimates rely on AE waveform averaging from multiple sensors, without accounting for rupture directivity. Given the small source dimensions, I suspect the rupture directivity could affect waveform characteristics.

9. Line 97, the "topographic gap" is unclear when first read, although it is explained in the Supplementary. It would be better to explain this when it first appeared in the main text.

10. Supplementary Fig. S1a, the labels for "triaxial strain gauge" and "biaxial strain gauge" are mistakenly written.

(Remarks on code availability)

Reviewer #3

(Remarks to the Author)

(Remarks on code availability)

Version 1:

Reviewer comments:

Reviewer #1

(Remarks to the Author)

The authors made significant efforts in replying to all my comments and investigating the many tests I proposed in my first round of review. I also appreciate that the new version now mentioned the proposed model as one potential physical model among others. I thereby consider that the new version of the manuscript is now ready for publication.

(Remarks on code availability)

All the details of the code are fine. I was just wondering because the authors mentioned a modified version of OpenSwpc to take into account the specific boundary condition of the experiment, is this modified version of this software available somewhere?

Reviewer #2

(Remarks to the Author)

The paper is well-written, and the data are carefully prepared. This is a revision, and I believe all my review comments have been appropriately addressed. Therefore, I support the publication of the manuscript in the current format.

(Remarks on code availability)

Reviewer #3

(Remarks to the Author)

The authors have appropriately addressed most reviewer concerns, except for the two following points.

First, if the minimum resolvable source duration is approximately $2.0 \mu\text{s}$, the authors should justify whether the truncated durations of events D118 and D126 affect the inferred scaling. If the true durations of these two events are allowed to be shorter, the scaling slope would be expected to exceed 0.04.

Second, the manuscript uses “dynamic source model” for the experiments and “dynamic rupture model” for the simulations, which is somewhat confusing. It is more common to use dynamic rupture model to refer to numerical simulations. But it is not a major issue.

(Remarks on code availability)

Response to reviewers on the manuscript: Dynamics of non-self-similar earthquakes illuminated by a controlled fault asperity

Kurama Okubo, Futoshi Yamashita, Eiichi Fukuyama
(NCOMMS-25-32014A)

February 6, 2026

We thank the Editor, the Associate Editor, and the three reviewers for their decision and for the thoughtful comments on the manuscript. We conducted additional analyses and revised the manuscript in response to their suggestions, which were helpful in improving its overall clarity and consistency. Below, we provide point-by-point responses together with the corresponding updates made in the revised manuscript. Black text indicates the original reviewer comments, **blue text represents our responses**, and **red text shows the revised text** included in the manuscript. Line numbers in the revised text correspond to those in the version with track changes. The source code used for the additional analyses presented in this rebuttal letter has been uploaded to the following GitHub repository:

https://github.com/kura-okubo/4mNonSelfSim_Paper/tree/dev/Others/Revision_Analysis

Comments by Reviewer #1

This article investigates the scaling of the source parameters of generated earthquakes during a laboratory experiment analog to a fault frictional slipping process. The authors build upon an already existing setup in order to record generated acoustic signal originating from some designed irregularities over the fault plane that have distinct frictional property. They identified from the recorded acoustic signals that the variation of moment of the recorded ruptures span a large range of moment while at the same time the rupture duration is almost constant. The authors propose that these results contradict the scaling predicted from the classical crack model where the rupture duration should vary with the moment of the recorded rupture. They propose a dynamic rupture model in order to reproduce the observed results and to suggest a possible physical origin for the observed non self-similar scaling.

I'm very impressed by the quality and precision with which the experiments have been carried out. The experimental set-up is clearly exceptional, with dense, cutting-edge instrumentation that makes it possible to record ruptures in a very convincing way. The care taken with the corrections to the measurements (sensor calibration, coupling coefficient, attenuation measurements) is clearly remarkable and lends real credence to the results presented in this work. The problem studied here is highly relevant and represents a new piece of information on the question regarding the existence or not of non-self similar seismic ruptures. As the authors stated, numerous recent measurements carried out in natural context suggest that these unusual ruptures do exist, but there is still some doubt as to the reliability of these findings due to the difficulty of fully understanding the role of attenuation in these results. This experiment therefore provides an unequivocal answer to the question of whether non-self-similar ruptures are possible. The article is well-written, and

the experimental observations are well presented such that the results seems very compelling. However, I'm a little less convinced by the numerical modelling part because it seems to me that a lot of parameters and hypotheses are imposed in a rather ad hoc way, although I have to admit I have less expertise on this part.

Finally, I just think it would have been great if the authors use patches with different sizes. This would have made it possible to validate that on the whole the cubic self-similar model is well represented, but also to validate that non-self-similar ruptures appear on different scales. I understand that this would have required a non-negligible piece of work. Therefore I would be looking forward for these new experiments.

Thank you very much for the concise summary and for clearly pointing out the weaker part of the manuscript. We have revised the discussion so that we do not present the model as a definitive explanation, but rather as one possible mechanism that is at least consistent with the experimentally inferred conditions for non-self-similar earthquakes. We now describe it more cautiously as a potential interpretation that may help to understand similar behavior in nature.

We agree that testing gouge patches with different sizes is an important next step. We have already begun preparing a new set of experiments using patches with varying dimensions, and we expect that these measurements will allow us to validate both the cubic self-similar scaling and the emergence of non-self-similar ruptures across different scales.

We have modified the paragraph about the perspective using larger patches as follows:

L510: Experiments using larger GPs will be crucial for further validating the source mechanisms of non-self-similar earthquakes. Although the observed scaling lies above the measurement limit on source duration, imposed by the water-level effect during attenuation deconvolution, larger patches would produce longer source durations that can be clearly separated from this limitation and would also clarify whether partial ruptures occur within the patch. Larger patches would additionally allow the nucleation location to be constrained by analyzing P-wave travel times, providing further constraints on the dynamic rupture modeling.

General comments:

1. Dynamic rupture model – As I understand here the authors propose that variation of the peak friction on the same fault patch can occur between successive ruptures of the same fault patch. The change of this peak friction which controls the magnitude of the rupture therefore provides an explanation for the variable moment observed in the experiments. In my opinion this is a bit the weakest part of this manuscript as actually the authors have no data that corroborate this hypothesis related to the change of the peak friction. The Fig S8 b is not totally convincing as a large scatter is visible. While this model is possible and the results of the model are well analyzed, I'm still a little unsure if this model actually captures what happens on the asperity and wonder if other explanations are possible to explain this lack of self-similarity.

Thank you for pointing out the speculative nature of the dynamic rupture modeling. We have carefully revised the manuscript to clarify the scope of our modeling approach, taking into account the feedback from the other reviewers as well. In the revised version, we explicitly state that the dynamic rupture model based on varying stress drop and self-healing friction is presented as one feasible explanation that complements other possible mechanisms for the non-self-similar scaling, rather than as the most probable mechanism for the GP events.

Regarding the concerns about the hypothesis of variable peak friction, we agree that the uncertainty in the cumulative local slip measurements relative to the inferred seismic moment prevents us from determining whether peak friction truly varied between events. In the revised manuscript, we clarify how this modeling assumption is incorporated into the construction of the dynamic rupture model, and emphasize that our analysis of cumulative slip relative to event size is intended only to demonstrate that this assumption is compatible with the observed trend, rather than to conclude that such variation occurred.

We have revised the description of the model construction and its underlying assumptions in the main text as follows:

L293: The final rupture size can, in principle, be constrained through kinematic inversion of static strain or slip changes (e.g., Dublanchet et al., 2024). However, no analyzable static offsets associated with the GP events are detected in either the strain or slip records, likely because the signals fall below the resolution limit of the measurements. While the final rupture extent remains unconstrained by static inversion, we assume that the source patch that produces positive stress drop coincides with the GP dimension, and that the final rupture extent is governed by the rupture dynamics under the prescribed barrier conditions.

Under this fixed-source-patch assumption, variations in stress drop are considered as a potential mechanism for the observed differences in seismic moment. We assumed different peak frictions within the patch under the condition that the strength excess and the residual friction level are comparable across all GP events to be modeled so that the stress drop varies. We then determined the best-fitting peak friction values that reproduces the observed source parameters exhibiting non-self-similar scaling as detailed in the Methods section.

To investigate whether the assumption of variable peak friction is compatible with the observations, we compared the seismic moment of GP foreshocks with the local cumulative slip, defined as the slip surrounding the GP accumulated from the initiation of preslip to the onset of the GP event, inferred from spatiotemporal interpolation of gap-sensor data (Fig. S10a^{*}). This local cumulative slip can be interpreted as an analogy for slip deficit on the GP (e.g., Johnson & Nadeau, 2002). The comparison showed that foreshocks with larger cumulative slip tend to exhibit larger seismic moments (Fig. S10b^{*}). Under our modeling assumption that the seismic moment scales with peak friction, the observed correlation between local cumulative slip and seismic moment does not contradict variability in peak friction among GP events.

*Formerly Fig. S8 in the original version.

2. To continue on my previous comment, it would have been very interesting, given the amount and the quality of data acquired during these experiments, to link the variation of the observed stress drop with the recorded parameters; local slip speed, normal stress, shear stress, total slip (not only the preslip), etc.. This would have been very informative and could have revealed what measurable physical parameter controls this variation of the stress drop.

Thank you for the suggestion. Here we have performed the comparison between the seismic moment M_0 of the foreshocks and aftershocks of GP events, and the recorded parameters, including local slip velocity, normal stress, macroscopic stress drop, slip on a main shock of stick-slip events, hold time and the total slip accumulated through the stick-slip cycles.

We focused on M_0 rather than stress drop because it is directly observable from the AE measurements. Under our modeling assumption that stress drop correlates with the seismic moment for a fixed source patch, M_0 can therefore be used as a proxy for variations in stress drop.

These comparisons are summarized in Fig. R1, which has been added as Fig. S11 in the Supplementary Material. Below, we outline the interpretation of each comparison.

1. Local slip velocity

We first compared M_0 to the local slip velocity. This relationship attracts an interest because Yamashita et al. (2022) presented the positive correlation between the local slip velocity and the foreshock's magnitude, which implies the effect of local slip velocity in the rupture mechanisms on the patch. We estimated the local slip velocity for each GP event by a linear interpolation of two gap sensor records near the GP. The slip speed was obtained at the timing of the onset of GP event through the linear interpolation with time.

Fig. R1a shows their comparison for the foreshocks and aftershocks of GP events. Most of the events are generated with the local slip velocity greater than ~ 1 mm/s, which is consistent to the argument of Yamashita et al. (2022) that the rupture on the patch can be dynamic with a certain level of local slip velocity. On the other hand, the correlation between M_0 and local slip velocity was not clear in our experiment. This may imply that the local slip velocity is not the primary controlling parameter for the seismic moment.

2. Local normal stress

We then examined the variations in local normal stress. Although the flat-jack pressures that generate the normal load are mechanically maintained at constant values, the local normal stress surrounding the GP can be slightly perturbed during each stick–slip cycle due to the dynamics of the preslip and the main shock. To quantify this effect, we analyzed the normal-stress component recorded by the nearby triaxial strain gauges (SGT07 and SGT23) installed on side surfaces of the rock specimen and averaged the measurements over a ± 1 ms window centered on the onset of each GP event.

As shown in Fig. R1b, the resulting distribution does not exhibit a clear correlation between local normal stress and seismic moment. This outcome is likely attributable to the very limited variation in the local normal stress under the present experimental conditions. Future experiments performed under a broader range of imposed normal stresses, for example 4–6 MPa, will be necessary to clarify whether M_0 depends on the local normal stress.

3. Macroscopic shear stress drop

We evaluated the general characteristics of the stick–slip events that ruptured the entire fault, including nucleation location, preslip rupture velocity, macroscopic stress drop, and coseismic slip (Fig. R2). The macroscopic shear stress (Fig. R2d) was calculated as the shear load measured by the load cell divided by the nominal fault area ($4 \text{ m} \times 0.1 \text{ m}$). We quantified the change in macroscopic shear stress immediately before and after each main stick–slip event and defined this difference as the macroscopic stress drop. This allowed us to examine whether the seismic moment of the GP events correlates with the overall stress drop of the corresponding stick–slip cycle.

The comparison reveals a scattered relationship between the two quantities (Fig. R1c). This suggests that the macroscopic shear stress drop is unlikely to control the seismic moment of the GP events.

4. Total slip of each stick-slip event

Fig. R1d compares the seismic moment of the GP events with the total slip of each stick-slip event, evaluated as the mean slip recorded by the 16 gap sensors (Fig. R2e). Each GP event is associated with the stick-slip event in which it occurred. The scattered data shows that the seismic moment of the GP events is not controlled by the total slip of the corresponding stick-slip event.

5. Hold time

We examined the hold time, defined as the duration between the previous stick-slip event and the onset of GP foreshocks, to assess whether time-dependent frictional healing influences the variation in stress drop. Fig. R1e compares the seismic moment of the GP events with their corresponding hold times. No systematic correlation is observed, indicating that time-dependent healing is unlikely to play a significant role in controlling the seismic moment of the GP events.

6. Total macroscopic slip

We examined the relationship between M_0 and the total macroscopic slip, defined as the cumulative sum of coseismic slip over successive stick-slip cycles, which grows to approximately 2 mm over the experiment. Each GP event is associated with the stick-slip event in which it occurred.

In contrast to the previous comparisons, the results suggest that smaller events tend to occur during the earlier stages of the experiment, when the cumulative slip is low, whereas larger events become more common at later stages as shown in Fig. R1f. This pattern may reflect the progressive evolution of the physical state of the gouge patch as slip accumulates, which could in turn influence the seismic moment of the GP events.

Unfortunately, we did not perform post-experiment microscopic analyses of the gouge material, so the evolution of the physical state of the patch cannot be directly assessed. We consider this an important direction for future work.

In summary, the comparisons showed that the total slip, which may reflect the evolution of the GP during the experiment, may play a role in controlling the seismic moment, whereas the other parameters considered here do not exhibit a systematic influence. Our perspective includes an advanced measurement which can more directly quantify the stress and slip evolution on the GP to identify the rupture mechanisms.

A description of the analysis metric is provided in Supplementary Note S2.

Supplementary Note S2: Comparison of seismic moment of GP events with recorded physical parameters

We examined how the seismic moment M_0 of the GP events relates to macroscopic and local measurements of physical parameters, in order to assess whether any of these quantities provide insight into the rupture mechanisms on the GP and help explain the observed variation in M_0 (Fig. S11). Below, we describe the metrics used to quantify each parameter from the macroscopic and local measurements. Hereafter, we use the term stick-slip events to denote ruptures involving the entire laboratory fault, in contrast to ruptures confined to the GP.

S2.1: Local slip velocity

The local slip velocity for each GP event was estimated by linearly interpolating two gap-sensor records located near the GP. The slip velocity at the GP onset time was computed from this interpolation procedure, following the approach of Yamashita et al. (2022).

S2.2: Local normal stress

Although the flat-jack pressures generating the normal load are mechanically maintained at constant levels, the local normal stress surrounding the GP can be slightly perturbed during each stick–slip event due to the dynamics of preslip and main shock. To capture this effect, we evaluated the normal-stress component recorded by the triaxial strain gauges near P3 (SGT07 and SGT23) installed on the side surfaces of the rock specimen, averaging the measurements over a ± 1 ms window centered on the onset of each GP event.

S2.3: Macroscopic shear stress drop

The macroscopic shear stress was computed as the shear load measured by the load cell divided by the nominal fault area ($4 \text{ m} \times 0.1 \text{ m}$). The macroscopic shear stress drop for each stick–slip event was then defined as the difference between the shear stress immediately before and after the main shock of the stick-slip event.

S2.4: Coseismic slip of each stick–slip event

The coseismic slip of each stick–slip event was evaluated as the mean slip recorded by the 16 gap sensors. Each GP event was associated with the stick–slip event during which it occurred.

S2.5: Hold time

To investigate whether time-dependent frictional healing influences variations in the seismic moment of GP events, we analyzed the hold time, defined as the duration between the preceding stick–slip event and the onset of GP foreshocks.

S2.6: Total macroscopic slip

The total macroscopic slip was computed as the cumulative sum of the coseismic slip over successive stick–slip events, eventually reaching approximately 2 mm during the experiment. Each GP event was linked to the cumulative slip value corresponding to the stick–slip event in which it occurred.

We have also clarified this analysis in the main text as follows:

L. 327: We also examined correlations between the observed M_0 and a set of local and macroscopic measurements, including local slip velocity, macroscopic stress drop, and hold time, to identify factors controlling variation in M_0 . All evaluated quantities are described in Note S2 and summarized in Fig. S11.

A possible trend is observed for the total macroscopic accumulated slip, in which smaller events tend to occur at earlier stages of the experiment, while larger events become more frequent at later stages (Fig. S11f). This tendency may reflect progressive evolution of the physical state of the gouge patch as macroscopic slip accumulates on the fault. However, none of the other additionally examined measurements show a clear correlation with M_0 . Consequently, these quantities do not

provide additional constraints that can be incorporated into the development of the dynamic rupture framework for the GP events.

Fig. R1 Relationships between the seismic moment (M_0) of GP events and various recorded physical parameters. a, Local slip velocity. Circles and triangles denote foreshocks and aftershocks, respectively. Horizontal error bars represent the standard error of M_0 estimated from the four AE sensors near the GP. b, Local normal stress. c, Macroscopic shear stress drop. d,

Coseismic slip of the associated stick–slip event. **e**, Hold time, defined as the interval between the previous stick–slip event and the onset of the GP foreshock. **f**, Total slip over successive stick–slip events.

Fig. R2 Overview of the stick–slip experiments. a, Macroscopic friction coefficient, computed as the ratio of the shear load recorded by the load cells to the normal load averaged over the eight flat jacks. **b**, Nucleation locations of the preslip. The labels E, W, and B denote preslip initiation on the east side, west side, and both sides of the fault, respectively. **c**, Preslip rupture velocity. Horizontal lines indicate the mean rupture velocities for preslip events nucleating on the east and west sides. **d**, Macroscopic shear stress drop, estimated as the difference in macroscopic shear stress immediately before and after each mainshock. **e**, Total slip for each event, measured from the onset of preslip to the time when all sections of the fault come to rest. Markers and error bars show the mean and standard deviation of slip obtained from the 16 gap sensors.

3. Could you show the spectra of at least one of the events with a low rupture duration to compare with an event of same moment but with larger rupture duration (for example D118 and M24). This should be quite informative for the reader to show that you can highlight the difference between these two spectra.

In response to this comment, we analyzed the spectra of two representative GP events with comparable seismic moments but different source durations, D118 and D24, to directly assess whether their spectral differences can be resolved. Event IDs have been unified using the prefix

“D” in the revised manuscript (see the Addendum of this rebuttal letter). Based on the source time function (STF) fitting, the estimated source parameters (M_0 [J], T_w [μ s]) are (0.08, 2.0) for D118 and (0.07, 2.5) for D24. The same spectral processing procedure as described in Supplementary Note S1 was applied.

Two sensors, AS23 and AS08, show higher corner frequencies for D118 (Figs. R3a and R3c). In contrast, the other two sensors (AS07 and AS22) exhibit only subtle differences (Figs. R3b and R3d). This reduced contrast may reflect sensor-dependent path effects between the source and individual sensors. Such effects could arise if the shorter-duration event (D118) corresponds to a partial rupture confined within the patch, which would lead to different propagation paths to individual sensors.

The stacked spectral ratio (Fig. R3e) departs from a flat spectrum, suggesting a potential distinction between the two source durations. We acknowledge, however, that the spectral differences, while detectable, remain modest. A larger patch (e.g., 15 mm in diameter) would shift the corner frequency to approximately 150 kHz, thereby improving the separation from the minimum resolvable source duration.

Fig. R3. Spectral analysis for two GP events exhibiting similar seismic moments but different source durations. a–d, Amplitude spectra of the P-wave windows recorded by the four AE sensors located near P3. The black and red lines correspond to the spectra of the event with the longer source duration (D24) and the shorter one (D118), respectively. The lower panels in each subplot show amplitude spectra normalized at 0.3 MHz. **e,** Stacked spectral ratio computed as the mean across the four AE sensors, also normalized at 0.3 MHz.

4. Most of the analysis and conclusion are drawn from ruptures on patch P3. What about ruptures on the other patches. Do they also show the same behavior as the one reported here?

Yes. Ruptures on the other gouge patches also exhibit signatures consistent with non-self-similar behavior. For example, we analyzed 16 foreshock and aftershock events associated with patch P5. As shown in Fig. R4, the recorded AE waveforms show high coherence, whereas the estimated seismic moments vary across a certain range. Although we have not yet performed the full source-parameter analysis for P5, such as attenuation correction and STF inversion as conducted for P3, the observed waveform similarity and stable pulse durations indicate that non-self-similar scaling is likely present on this patch as well. We therefore expect that the scaling behavior identified on P3 is not specific to a single gouge patch but may represent a more general feature of rupture processes on isolated asperities within the laboratory fault.

We added this figure as Fig. S9, and explained this in the main text as follows:

L265: We also analyzed the foreshocks and aftershocks generated from the secondarily activated patch P5 (Fig. S9). The recorded waveforms show high phase coherence, whereas their amplitudes vary across a certain range. While we did not conduct a full source analysis for these events, the observed characteristics indicate that the non-self-similar behavior is not unique to P3 but may be a robust feature of ruptures on the GPs.

Fig. R4 Observed non-self-similarity at patch P5. a, AE waveforms recorded at AS27 for 16 events generated on gouge patch P5 ($x = 2.75$ m). a, Seismic moments were estimated through preliminary waveform fitting using synthesized Green's functions. The same preprocessing as in Fig. 1c of the main text was applied to these waveforms. b, Normalized P-wave arrivals. c,

Superposition of normalized P-waveforms, highlighting the similarity in source duration across events of different magnitudes.

Specific comments

L35: ‘classical source model’, you mean ‘shear crack model’

Thank you for the clarification. We modified the text as follows:

L42: **Classic source models based on a shear crack (e.g., Madariaga, 1976),**

In the introduction you could also mention the article of Tsai and Hirth, 2019, which propose that the non-self-similar ruptures can be explained with a completely different model than the shear crack model (particle impacts). As you know fault gouge particle size you could also see if your results fit this model prediction. As this particle impact model is maybe not the most standard model proposed for the source scaling, I let the authors decide whether or not this analysis is worthwhile.

Thank you for the suggestion regarding the particle-impact model proposed by Tsai and Hirth (2020). We evaluated its characteristic time scale using their approximation

$$T_c \approx 0.014R, \tag{R1}$$

where T_c is the impact time scale and R is the effective particle radius. Following Text S3 of Tsai and Hirth (2020), we used the 73rd percentile particle size R_{73} . From the particle size distribution of the gouge used in the patch (Figure S12 in the manuscript), we obtained $R_{73} = 7.3 \mu\text{m}$. This gives $T_c = 0.10 \mu\text{s}$, corresponding to a characteristic frequency of about 10 MHz.

This frequency is much higher than the characteristic source duration of about $2.5 \mu\text{s}$ for the non-self-similar GP events. In addition, while such high-frequency components may exist at the source, they would be strongly attenuated and fall outside the measurement range of our AE sensors. Therefore, the particle-impact mechanism is unlikely to control the non-self-similar scaling observed in our experiments.

Given that this mechanism does not affect our analysis of the dynamic rupture mechanisms of non-self-similar events, we decided to omit the particle-impact model from the revised manuscript. We appreciate the reviewer’s suggestion, and note that it remains a possible alternative framework for non-self-similar scaling. We acknowledge that this mechanism should be considered when analyzing high-frequency components in source spectra.

L103: stick slip event: you mean system size stick slip event? Or could it be just local ones?

It indicates the rupture of system size. We modified the text as follows:

L144: **We observed 56 stick-slip events that ruptured the entire fault during a single experimental run.**

L161: four closest sensors? did you try using the eight closest sensors? Why this choice. Is the signal becoming rapidly too weak or attenuation becomes significant?

The use of the four closest sensors reflects a trade-off: increasing the number of sensors generally improves the statistical robustness of the moment–duration estimates and reduces the influence of

directivity, but adding more distant sensors also increases the bias from low signal-to-noise (S/N) ratio and path effects.

Although the S/N ratio remains acceptable at more distant sensors for sufficiently large events, where P-wave pulses are still identifiable (Fig. 2c), these pulses are not reliably resolved at distant sensors for smaller GP events. As a result, including such sensors would introduce magnitude-dependent biases and hinder a fair comparison between large and small events.

We have added the following text to clarify this point in the manuscript:

L206: We excluded the farther sensors from the analysis because the signal-to-noise ratio is low, particularly for smaller GP events, which limits a fair comparison across events with different magnitudes.

L251-252: – does this mean this mechanism is specific to fault gouge experiment?

Thank you for the clarification. We did not intend to imply that the non-self-similarity promoted by self-healing friction occurs only in the presence of fault gouge. Our model is based on self-healing friction, which can arise under a range of frictional constitutive laws that incorporate healing effects, including rate-and-state friction laws.

We modified the text as follows:

L386: From a mechanical perspective, self-healing is expected under velocity-dependent friction laws (e.g., Perrin et al., 1995) and has been documented in rock friction experiments (e.g., Sone and Shimamoto, 2009; Fukuyama and Mizoguchi, 2010; Goldsby and Tullis, 2011), providing a relevant analogue to natural fault conditions. In our implementation, this effect is represented by introducing a linear slip-strengthening part after a prescribed slip distance D_s , which limits further slip (Methods).

L263: – ‘ D_c vary with magnitude of the event’, do you mean the size of the patch? It is not clear to me why D_c should scale with magnitude (for constant size events).

We do not vary D_c with patch size, which is fixed in our models. Instead, under fixed-source-patch assumption, we vary D_c with event magnitude, equivalently with stress drop, in order to control the moment acceleration of STFs and to reproduce the observed seismic moment M_0 and source duration T_w . Below we provide the nondimensional reasoning that clarifies how stress drop and D_c jointly control the STF characteristics.

In dynamic rupture models, the slip velocity can be scaled as

$$\dot{\delta}(t) \sim \frac{\Delta\sigma c_s}{G},$$

(e.g., Dunham and Bhat, 2008), where $\Delta\sigma$ is stress drop, c_s is shear wave speed, and G is shear modulus.

The characteristic timescale of the rupture process is given by R_0/c_s , where R_0 is the process zone size near the rupture tip (Palmer and Rice, 1973). Normalizing the slip velocity scale by this timescale yields the characteristic slip-acceleration scale,

$$\ddot{\delta}(t) \sim \frac{\Delta\sigma c_s^2}{GR_0}.$$

Because R_0 scales with D_c , this leads to

$$\ddot{M}_0(t) \propto \ddot{\delta}(t) \propto \frac{\Delta\sigma}{D_c},$$

For the target GP events with variable stress drop, keeping D_c constant causes excessively large moment accelerations, resulting in the modeled STF that cannot reproduce the observed seismic moments and source durations. We therefore allow D_c to increase with event magnitude following Eq. (10) in the main text to moderate the moment acceleration and enable quantitative agreement with the observed source parameters.

We added the following sentence to the manuscript to clarify this point:

L404: We allowed the characteristic slip distance of the slip-weakening law, D_c , to increase with the magnitude of the target events in order to moderate the stress-drop-dependent moment accelerations in the modeled STFs and to achieve quantitative agreement with the observed source parameters. Details of the systematic approach for model parameter selection are provided in the Methods section.

L293-294: Maybe I would avoid this comment that is too general. For example, if you have borehole measurement close to a fault you will be in very similar condition as in laboratory experiment (or maybe even better), regarding the attenuation effect. I think a real strength of these laboratory experiments is that you can control and record many parameters during the process (strain, slip, etc...).

I agree that the original sentence placed too much emphasis on the advantage of laboratory experiments over natural observations. We have toned down this statement as follows:

L457: The precision of these corrections shows ~~a significant~~ an advantage of laboratory experiments, particularly in settings where attenuation effects can be independently constrained ~~over natural observations~~.

L307-314: I am not sure to understand very well what you really mean here. It seems that you try to support the variation of peak friction between successive stick-slip from a physical basis but it remains quite speculative.

The paragraph on micromechanisms was intended to outline plausible physical processes that could account for the variation in stress drop assumed in our proposed dynamic rupture model, rather than to provide definitive support for this modeling strategy.

To clarify this intent and avoid overinterpretation, we have revised the text as follows:

L479: Given the variation of peak friction required in the proposed dynamic rupture model, the potential micromechanisms may originate from differences in the formation of force chains (Sammis & Steacy, 1994; Anthony & Marone, 2005; Gao et al., 2018), or from differences in slip interfaces within the gouge layer or at the boundary between gouge and host rock.

In addition, we removed the previous sentence regarding shear localization to avoid overstating a specific mechanism based on the current modeling framework.

Comments by Reviewer #2

This paper conducts laboratory experiments to investigate non-self-similar earthquakes, where an earthquake's source duration remains nearly constant regardless of its seismic moment, diverging from typical earthquake scaling laws. The authors used a meter-scale laboratory fault with controlled gouge patches to generate microearthquakes exhibiting this behavior. Through careful sensor calibration and attenuation correction, they developed a dynamic rupture model that attributes this non-self-similarity to a combination of variable stress drop and self-healing friction on isolated asperities, suggesting it can occur in diverse tectonic settings beyond previously assumed strong rupture barriers. The study highlights the advantages of laboratory experiments for precise measurements in understanding earthquake mechanisms, addressing a critical question in earthquake physics and allowing for a more accurate estimation of scaling laws and source mechanisms in earthquakes. Here are several suggestions that could further enhance the clarity of the paper.

Thank you very much for the summary and the thoughtful comments. We hope that our responses below address the comments and further improve the clarity of the revised manuscript.

1. The paper provides a clear definition of non-self-similar earthquakes as "clusters of events that show similar waveforms in phase but vary in amplitude". They then state that while these "may be categorized as repeating earthquakes, the latter term also encompasses events with nearly identical phase and amplitude". However, throughout the Discussion, these terms are often used in close conjunction or somewhat interchangeably (e.g., "variation in the source parameters within the cluster is likely analogous to that observed in the natural repeating earthquakes"; "insights into the generation processes of the repeating earthquakes"; "as has been done in studies of the repeating earthquakes"). Please explicitly state if the non-self-similar events generated in your experiments are considered a subset of repeating earthquakes or if "repeating earthquakes" is primarily used as a broader, less specific term for events with high waveform similarity.

Thank you for the suggestion. We agree that the definition of non-self-similar earthquakes should be stated more clearly and that the terminology should be used consistently throughout the manuscript. First, we define the non-self-similar earthquakes as clusters of events generated by a non-self-similar source process. When such a cluster shares a common source location, the waveform phases are expected to be coherent. In this study, we focus on this type of cluster.

In our context, we consider non-self-similar earthquakes that share a common source location to be a subset of repeating earthquakes (Fig. R5). Within this definition, events that exhibit uniform amplitude and source duration are also categorized as repeating earthquakes, but they fall outside the non-self-similar subset.

We acknowledge an alternative framework in which repeating earthquakes are defined as events with both nearly identical amplitudes and source durations, and are therefore treated as a subset of non-self-similar events. However, we adopt the former interpretation, in which non-self-similar earthquakes are treated as a subset of repeating earthquakes, because events with coherent waveforms but variable amplitudes are commonly regarded as repeating earthquakes in observational studies (e.g., Nadeau et al., 1995).

We have added Fig. R5 as Supplementary Figure S1 and revised the corresponding text in the manuscript to improve the consistency of the terminology, as follows:

L60: In this study, we define non-self-similar earthquakes as clusters of events generated by a non-self-similar source process, as illustrated in Fig. 1a. We focus on clusters in which events originate from a common source location and show highly coherent waveform phases but variable amplitudes. Clusters that exhibit deviations from self-similar behavior less pronounced than those considered in the present analysis (e.g., Trugman, 2020) are not examined.

Under this framework, non-self-similar earthquakes that share a common source location are treated as a subset of repeating earthquakes, without considering recurrence intervals as a classification criterion. Consequently, repeating earthquakes include non-self-similar clusters and the complementary clusters composed of events with nearly identical waveforms in both phase and amplitude. This classification follows common practice in observational seismology, where events with coherent waveforms are typically classified as repeating earthquakes even when modest amplitude variations are present (e.g., Nadeau et al., 1995). Our framework aligns with established observational conventions while explicitly distinguishing non-self-similar behavior within repeating earthquake clusters.

Fig. R5 Schematic illustration of the classification of non-self-similar earthquakes. Non-self-similar earthquakes are defined as events generated by a non-self-similar source process. This study focuses on clusters of non-self-similar earthquakes that share a common source location, and therefore exhibit high waveform coherence. Within this framework, repeating earthquakes are characterized by a common source location and include both non-self-similar clusters and complementary clusters exhibiting nearly identical waveforms in both phase and amplitude.

2. The authors use a slip-weakening friction law with self-healing to conduct dynamic rupture modeling. The simulated events maintain constant source duration. Based on the simulation, they claim that variable stress drop, achieved by setting different peak frictions within the gouge patch, is crucial for varying seismic moments, and also reveal that the self-healing friction is "key mechanism for reproducing non-self-similar scaling" by preventing excessive slip. First, why is the slip-weakening plus self-healing law used?

The slip-weakening friction law with an added self-healing term was adopted because it provides a minimal extension to the conventional slip-weakening framework that can reasonably reproduce the observed non-self-similar scaling. As described in L358–368 of the main text, our preliminary simulations using only the slip-weakening law resulted in self-similar behavior: in the absence of a substantial rupture barrier, the rupture area naturally expands with increasing stress drop. As a result, the slip is accumulated excessively in the center of patch, which elongates source duration and causes it to scale with the seismic moment.

In principle, non-self-similar behavior could also be achieved by varying the nucleation location within the patch or prescribing different rupture extents in the barrier region. We cannot exclude these mechanisms as potential explanations for the observed non-self-similar events, because the measurement resolution is insufficient to determine the nucleation location from the far-field P-wave onset or to infer the final rupture size from static strain changes (see also our reply to Major Comment #4 of Reviewer #3). However, implementing those approaches would require thorough tuning of the source characteristics, which we think would undermine the generality of the model in reproducing non-self-similar scaling.

Instead, we explored incorporating a self-healing mechanism, which represents a minimal modification capable of reproducing the observed non-self-similar behavior. Self-healing friction has been demonstrated in rock friction experiments (e.g., Sone and Shimamoto, 2009; Fukuyama and Mizoguchi, 2010; Goldsby and Tullis, 2011). In addition, prior studies (e.g., Wang and Day, 2017; Kano et al., 2025) have shown that pulse-like ruptures exhibit higher corner frequencies than crack-like ruptures under comparable rupture velocities, implying that healing can effectively shorten the source duration. Motivated by this insight, we incorporated a self-healing term into the slip-weakening law.

We still need to tune parameters, including the characteristic distance for self-healing D_s , to match the model to the observed source parameters of the GP events. Nevertheless, we consider self-healing, in combination with variable stress drop, to have the potential to robustly explain the non-self-similar behavior, because it limits excessive slip and thereby prevents the elongation of source duration as detailed in the revised manuscript (see also our reply to Comment #6 of Reviewer #2).

We revised the manuscript to detail this context as follows:

L358: We applied a linear slip-weakening friction law to the GP, instead of a rate-and-state formulation, to simplify the model setup. Under this formulation, the weak barrier allows the rupture to expand beyond the source region, analogous to rupture penetrating a velocity-weakening barrier (e.g., Molina-Ormazabal et al., 2023), particularly for events with large stress drops. Since both seismic moment and source duration increase with rupture area, as predicted by classical source models, this behavior is not compatible with the non-self-similar scaling (Fig. S12).

In principle, the observed source parameters could be matched by varying the nucleation location within the GP or by prescribing different final rupture extents. We do not exclude these scenarios; however, in the absence of sufficient measurement resolution to constrain the nucleation location from far-field P-wave onsets or to infer the final rupture size from static strain changes, such approaches would require substantial event-by-event tuning of source characteristics. In our view, this would limit the generality of the modeling strategy.

We therefore incorporated a self-healing friction term as an additional factor in the dynamic rupture model (Fig. 4a*). This mechanism suppresses excessive slip within the GP that would otherwise arise from rupture expansion into the weak barrier region, thereby reducing the elongation of the source duration. This behavior would be consistent with the analyses of Wang and Day (2017) and Kano et al. (2025), which demonstrated that pulse-like ruptures exhibit higher corner frequencies than crack-like ruptures under comparable rupture velocities. From a mechanical perspective, self-healing is expected under velocity-dependent friction laws (e.g., Perrin et al., 1995) and has been documented in rock friction experiments (e.g., Sone and Shimamoto, 2009; Fukuyama and Mizoguchi, 2010; Goldsby and Tullis, 2011), providing a relevant analogue to natural fault conditions. In our implementation, this effect is represented by introducing a linear slip-strengthening part after a prescribed slip distance D_s , which limits further slip (Methods).

* Formerly Fig. 4c in the original version.

Will the more comprehensive rate-state friction law give the same or similar results in terms of constant source duration and non-self-similar scaling?

We are indeed motivated to extend the analysis using a rate-and-state friction framework, as it can naturally produce self-healing behavior without prescribing it explicitly. Although we have not yet conducted full dynamic rupture simulations with rate-and-state friction, our expectation is that non-self-similar behavior could still emerge even without a strong rupture barrier, provided that the stress and frictional conditions within the source patch allow self-healing to develop.

To reflect this perspective, we have added the following statement to the manuscript:

L503: Further numerical investigation using a rate-and-state friction framework would be worthwhile, as self-healing behavior can arise naturally from the constitutive law (e.g., Perrin et al., 1995). Dynamic rupture models that include a velocity-weakening or velocity-neutral region surrounding the source patch, together with variations in stress drop within the patch, may provide additional support for the role of self-healing friction in the emergence of non-self-similar scaling in the absence of a strong rupture barrier.

Second, the simulations set the varying peak friction and self-healing as the input conditions and generate the non-self-healing phenomenon. Logically, this only means that the varying peak friction and self-healing are the causes of non-self-healing, but may not necessarily prove that the varying peak friction and self-healing are the fundamental mechanisms of non-self-healing. This seems a weak reasoning.

We agree that our simulations using varying peak friction and self-healing do not demonstrate these factors as the fundamental or universally governing mechanisms of non-self-similar earthquakes. Our intention is not to argue that this combination is the only, or the most broadly applicable, explanation. Rather, we present it as one plausible mechanism that is consistent with our experimental configuration, quantitatively explains the observed source parameters, and complements previously proposed models.

We revised the manuscript to clarify this point:

L25: We further develop a dynamic rupture model that quantitatively explains the observed source parameters by incorporating a fixed source-patch size, variable stress drop within the patch, and

self-healing friction. This modeling framework complements previously proposed models and expands the range of tectonic conditions under which non-self-similar earthquakes may occur.

L444: Overall, these results indicate that a dynamic rupture model incorporating variable stress drop on an isolated patch and self-healing friction provides one feasible, quantitative, and internally consistent explanation for the observed non-self-similar scaling on the GP events. This framework complements previously proposed mechanisms, and thereby expands the range of conditions under which non-self-similar scaling may emerge.

Please also find the reply to comment #3 below to emphasize this context in the main text.

3. The authors have summarized previous proposals for non-self-similar scaling, including "fixed source dimensions with variable stress drop or accelerating rupture velocity" and models "based on the isolated asperity patches". The authors' model clearly adopts the "former approach, modeling variations in stress drop". While they briefly contrast their model's lack of a "substantial rupture barrier" with some previous assumptions. Why does their combination of variable stress drop and self-healing friction provide a more robust or broadly applicable explanation for non-self-similar scaling, especially compared to other proposed mechanisms (e.g., rupture acceleration)?

Our intention is not to argue that the present model, combining variable stress drop and self-healing friction, is more representative or more robust than previously proposed mechanisms for explaining non-self-similar scaling. Rather, the purpose of our dynamic-rupture modeling is to demonstrate one additional plausible physical mechanism that can quantitatively reproduce the non-self-similar scaling observed in our laboratory experiments, thereby offering potential insight into natural fault behavior. Our view is that this approach broadens the range of conditions under which non-self-similar scaling can be reproduced, complementing the mechanisms proposed in earlier studies.

We have revised the manuscript, to clarify this modeling strategy and its intended scope as follows:

L272: To investigate the source mechanisms responsible for the non-self-similar scaling of the GP events, we developed a dynamic rupture model constrained by the estimated source parameters and the controlled GP configuration. Our aim is not to evaluate this model against previously proposed explanations (e.g., Lin and Lapusta, 2018; Cattania, 2023), nor to argue which mechanism is most representative of natural non-self-similar earthquakes. Instead, we construct a feasible dynamic rupture framework that is consistent with our experimental configuration, quantitatively explains the observed source parameters, and complements previously proposed models. Although some aspects remain incompletely constrained, including the final rupture extent and the shear-traction history on the GP, our objective is to identify the physical factors capable of robustly reproducing non-self-similar scaling in our experimental system and to provide insight into possible mechanisms applicable to natural earthquakes.

4. The manuscript describes a large-scale biaxial rock friction experiment in which the normal stress is spatially varied—set to 2.0 MPa at the center and reduced to 0.7 MPa at both edges of the simulated fault (Fig. 1b). The authors state that this reduction at the edges is to “promote the smooth nucleation of stick-slip events.” However, the rationale for specifically choosing 0.7 MPa as the edge stress value is unclear. Is this value based on prior empirical optimization, numerical modeling, or physical constraints of the setup?

We selected the edge normal stress as one-third of the macroscopic normal stress based on empirical optimization aimed at generating preslip on the fault (see the detailed explanation in the following response).

Furthermore, what exactly is meant by “smooth nucleation” in this context? How do you define “smooth nucleation” and how will this influence the results in the present paper?

Regarding the term “smooth nucleation”, our early experiments conducted under uniform normal loading frequently resulted in abrupt dynamic nucleation at the fault edges, producing large-amplitude AE signals. These signals inhibited the propagation of preslip that generates foreshocks and also masked the subsequent signals of interest. This behavior is consistent with the stress concentration at the specimen edges caused by elastic deformation, as demonstrated by the finite-element analysis of Yamashita et al. (2022).

To promote the preslip propagation, we reduced the normal stress at both edges. We found that reducing the normal stress to approximately one-third of the central value was sufficient to cause the preslip propagation along the fault. We emphasize that 0.7 MPa is not a unique or universal value; rather, some degree of reduction is required to avoid abrupt nucleation in this experimental configuration.

In this study, we defined “smooth nucleation” as nucleation preceded by measurable preslip propagation along the fault. This preslip is essential for analyzing the gouge-patch events, which are triggered by the surrounding slip. For clarity, we replaced the term with preslip propagation in the main text as follows:

L108: The decreased normal loading at the edges promotes preslip propagation from the fault edges prior to the stick-slip events.

5. Page 4, line 90, “We selected the size of the GP as a diameter of 8 mm, which was preferred as the fraction of the size to the simulated fault was small enough to mimic the source embedded in an infinite medium at depth, compatible with the natural condition.” The manuscript states that an asperity diameter of 8 mm was selected to mimic a buried seismic source in an infinite medium. However, the basis for this specific choice is unclear. Was it determined by empirical evidence, prior experiments, scaling analysis, or simply practical considerations? The term “preferred” is vague and lacks scientific justification.

We determined the size of the gouge patch based on the expected source spectrum, the critical nucleation size, and the required size fraction relative to the overall fault. First, we estimated the expected corner frequency of a GP event using Madariaga’s circular fault model, which yields a corner frequency of approximately 300 kHz for an 8 mm diameter. This falls within the frequency band over which the AE sensors were calibrated using the PZT source. Second, we estimated the critical nucleation length under the measured normal stress using pressure-sensitive film, obtaining values on the order of a few millimeters. This suggests that the patch should be several times larger than this nucleation size. Finally, we avoided choosing an excessively large patch (e.g., 50 mm in diameter on the 100 mm in width of the fault), as such a large asperity would be affected by the free surfaces at the specimen edges, potentially biasing rupture nucleation on the patch and the associated radiation pattern.

We have revised the explanation of the patch-size selection in the manuscript as follows:

L119: We selected the size of the GP to be 8 mm in diameter for the following reasons: (i) the expected corner frequency of a seismic source of this size is approximately 300 kHz based on the

classical shear crack model (e.g., Madariaga, 1976), which lies within the measurement range of our AE sensors (see Methods); (ii) the patch size must exceed the critical nucleation size, which we estimated to be on the order of a few millimeters based on approximate normal stress on the patch (see Methods) together with assumed frictional parameters of the slip-weakening law; and (iii) the ratio of the patch size to the overall simulated fault dimension is sufficiently small to approximate a buried source in an effectively infinite medium, consistent with natural settings.

6. The slip-weakening friction law shown in Fig. 4c has a very flat stable friction and a very sharp increase in self-healing. I usually see people using a smooth transition of friction in such a law, i.e., a smooth increase in healing. Will the sharp change in healing influence the results?

We performed additional dynamic rupture simulations using a smoothed self-healing friction law. The friction coefficient was prescribed as:

$$\mu(\delta) = \begin{cases} \mu_s - \frac{\delta}{D_c}(\mu_s - \mu_d), & (0 \leq \delta \leq D_c) \\ \mu_d, & (D_c < \delta \leq D_s) \\ \cosh(k_{heal}(\delta - D_s)) - 1 + \mu_d, & (D_s < \delta) \end{cases}$$

where μ_s and μ_d , are the static and dynamic friction coefficients for the slip-weakening law, D_c is the characteristic slip-weakening distance, D_s is the predefined slip distance at the onset of self-healing, and k_{heal} controls the smoothness of the self-healing. In the present tests we set $k_{heal} = 8 \times 10^6$. This modified slip-weakening law yields a much smoother increase in frictional strength during healing than the sharp self-healing adopted in the main text (Fig. R6a).

We applied this smoothed law to the largest target event, D129. The resulting STF is nearly identical to that obtained with the sharp self-healing implementation (Fig. R6b). The main difference appears in the spatial pattern of shear stress: with the smoothed law, the stress increase due to healing is more spatially distributed than in the sharp-healing case (Fig. R7).

We concluded that the non-self-similar behavior can still be reproduced even when a smooth self-healing law is used. This is because the key mechanism responsible for non-self-similar behavior is the suppression of the excessive slip at the patch center. The extent and magnitude of this slip suppression increase with the stress drop within the patch (Fig. R8), thereby contributing to the emergence of non-self-similar behavior. The next step is to perform dynamic rupture simulations based on a rate-and-state friction law, in which healing emerges from the constitutive behavior rather than from a prescribed smoothness.

We have modified the Fig. 4e to demonstrate the region where the slip is limited by self-healing friction as shown in Fig. R9. Fig. R8 is also included as Fig. S13 in the Supplementary Material. The paragraph to discuss this context is added in the main text as follows:

L430: The key mechanism responsible for the non-self-similar behavior is the suppression of excessive slip at the patch center by self-healing friction (Fig. 4e). In the absence of healing, slip growth localizes at the patch center, particularly under low barrier efficiency. This localization is suppressed once self-healing friction is activated, limiting further slip accumulation. In addition, under the assumption that the prescribed slip distance for the healing part, D_s , scales with D_c , the

extent and magnitude of slip suppression increase with the stress drop within the patch (Fig. S13), which in turn leads to tighter STF's for larger events. Deviations from the elliptic slip distribution predicted by the circular crack model have been reported by Wang and Day (2017). Kano et al. (2025) further showed that self-healing slip pulses can generate an additional high-frequency characteristic, reflected in double-corner frequencies, consistent with the physical interpretation of our modeling framework. Taken together, these effects result in nearly constant source durations across the target events, accounting for the observed non-self-similar behavior.

Fig. R6 Traction history and STF for the largest target event D129. a, Comparison of the traction history for smooth (red) and sharp (black) self-healing. **b,** Corresponding STF's.

Fig. R7 Snapshots of dynamic rupture simulations with a smoothed self-healing friction law. Slip velocity, slip, and shear stress change are shown for the D129 event. The cyan circle in the slip distribution panels delineates the region where slip is limited by the self-healing friction law.

Fig. R8 Comparison of slip and shear stress change for the target events. Each panel shows the final snapshot ($t = 4.0 \mu\text{s}$) of the dynamic rupture simulations with self-healing friction. $\Delta\tau$ denotes the stress drop within the gouge patch. The spatial extent of the region where self-healing is activated increases with seismic moment M_0 , which is equivalent to stress drop in the present framework.

Fig. R9 Modified version of Fig. 4. A contour delineating the region where self-healing friction is activated has been added to panel e, highlighting the spatial extent of slip suppression during rupture evolution. The order of panels a–c has been rearranged to align with the revised manuscript structure. Panel b additionally includes annotations identifying the source patch (PCH) and the surrounding outer region (OUT).

7. The magnitudes of the events generated are within the range of Mw -7.3 and -6.0. Will such a narrow magnitude range bias the obtained scaling?

We acknowledge that the magnitude range of the GP events is limited in the current experimental setup. To further evaluate the robustness of the scaling behavior, we are planning additional experiments using gouge patches of different sizes, which will enable us to extend the magnitude range and independently verify the observed scaling.

8. The source duration estimates rely on AE waveform averaging from multiple sensors, without accounting for rupture directivity. Given the small source dimensions, I suspect the rupture directivity could affect waveform characteristics.

To assess whether variations in rupture directivity across events introduce bias in the observed waveform characteristics, we examined the sensor-dependent measurements of seismic moment and source duration, as shown in Fig. R10. For each GP event, four independent measurements

were obtained from the closest AE sensors and used to compute the mean values and standard errors reported in Fig. 3c of the main text.

Although the measurements vary among sensors, likely due to rupture directivity and measurement uncertainties, the scatter is not large enough to affect the interpretation of the scaling behavior of the GP events. We therefore use the averaged values and their standard errors to discuss the overall scaling, noting that directivity-related variability does not systematically bias the inferred source characteristics.

Fig. R10 Measurements of seismic moment and source duration for individual events obtained from four AE sensors. Each panel shows the estimates of M_0 and T_w derived from the

four closest sensors to P3. The mean values are annotated in the upper left of each panel. Colored markers indicate measurements for the target GP event, whereas gray markers represent measurements for the other events.

9. Line 97, the “topographic gap” is unclear when first read, although it is explained in the Supplementary. It would be better to explain this when it first appeared in the main text.

Thank you for the clarification. We revised the text as follows:

L136: The GP region has a slightly higher topographic height than the surrounding fault surface. This topographic offset leads to a concentration of normal stress onto the GP when the upper rock specimen is stacked, which strengthens the frictional coupling.

10. Supplementary Fig. S1a, the labels for "triaxial strain gauge" and "biaxial strain gauge" are mistakenly written.

Thank you for pointing this out. We have corrected the labeling error in Fig. S2a (formerly Fig. S1a in the original version).

Comments by Reviewer #3

This manuscript presents interesting results from laboratory earthquake experiments with seven 8 mm-diameter circular gouge patches placed on a meter-scale fault interface. The authors propose that the laboratory earthquakes, nucleated within the gouge patches, show a non-self-similar scaling, i.e., the source duration weakly dependent on the seismic moment. The non-self-similar scaling was previously reported in natural faults. They use dynamic rupture simulations to reproduce the observed non-self-similarity, and suggest that a combination of variable stress drop and self-healing friction is needed to fit the observations, with the latter preventing rupture extending outside the gouge patch. The laboratory experiments in this study provide interesting new data that can help elucidate the mechanisms of small earthquakes. And these data may connect the scaling relation across different length scales from laboratory observations to larger natural earthquakes. I have several comments about this manuscript before it can be published.

Thank you very much for the constructive comments on the manuscript. They helped deepen parts of the discussions that were not fully addressed in the initial version. We hope that the responses below clarify the concerns you raised.

Major comments:

1. Did all the laboratory earthquakes rupture the entire gouge patch?

Thank you for pointing out the partial rupture on the patch. As noted in L495, we consider that several GP events, particularly those with small seismic moments and short source durations, may indeed represent partial ruptures.

If the smallest events only ruptured a small fraction of the gouge patch, can the current measurements detect and constrain them?

Our analysis indicates that the minimum resolvable source duration is approximately 2.0 μ s, limited by the water-level stabilization needed to deconvolve attenuation effects (see also our reply

to Major Comment #2 below). If rupture size directly controls the source duration, then partial ruptures producing durations shorter than $2.0 \mu\text{s}$ cannot be distinguished, because attenuation distorts the far-field P-wave displacement pulse and masks the true duration.

It seems that the dynamic rupture simulations have implicitly assumed the condition of a fixed rupture size.

While the source patch size that produces a positive stress drop is prescribed in our models, the final rupture size is not fixed a priori but emerges from the rupture dynamics under the imposed barrier conditions. As discussed in our response to Major Comment #4 below, the final rupture size cannot be reliably constrained by the existing side-surface strain and slip measurements. We therefore acknowledge that improved measurements are necessary to better constrain the dynamic rupture models.

We have revised the main text to explain this context as follows:

L293: The final rupture size can, in principle, be constrained through kinematic inversion of static strain or slip changes (e.g., Dublanchet et al., 2024). However, no analyzable static offsets associated with the GP events are detected in either the strain or slip records, likely because the signals fall below the resolution limit of the measurements. While the final rupture extent remains unconstrained by static inversion, we assume that the source patch producing positive stress drop coincides with the GP dimension, and that the final rupture extent is governed by the rupture dynamics under the prescribed barrier conditions.

I appreciate the authors' efforts to account for the attenuation effects and the precise calibration of the instruments to reduce possible biases, but I think a missing element is verifying whether the instruments can constrain the source duration of smaller events when their durations are much shorter than $2 \mu\text{s}$ (the smallest event).

The two smallest events (D118 and D126) likely fall within the resolution limit imposed by attenuation deconvolution. Events with true durations shorter than $2.0 \mu\text{s}$ would appear artificially elongated to $2.0 \mu\text{s}$ in the processed waveform, making it impossible to determine whether they represent small partial ruptures or smaller sources distorted by attenuation.

Importantly, all non-self-similar events analyzed in the main text have durations statistically longer than this limit, which supports the robustness of the scaling for the GP event cluster. We also acknowledge that future experiments using larger gouge patches will help further separate the physical source duration from the measurement limitation.

We added this context in the main text as follows:

L231: The two smaller events (D118 and D126) most likely reflect limitations inherent to the attenuation-deconvolution process with water-level stabilization. Under these conditions, sources with shorter durations cannot be resolved because attenuation distorts the far-field P-wave displacement pulse beyond the recoverable bandwidth.

Although Figure S17 has shown the ball-drop impact, its duration of $5 \mu\text{s}$ (found in Fig S17) is longer than that of the largest laboratory earthquake ($\sim 3 \mu\text{s}$), and therefore it does not directly verify the resolution for short source durations.

Thank you for the clarification. The impact duration of a ball-drop source scales with the square of the ball radius; we used a 2-mm ball, which is close to the practical minimum size for conducting

the test. For this reason, it is currently difficult to use ball-drop experiments to evaluate the resolution limit for very short source durations.

2. It may be helpful to show the Green's function of each AE with the attenuation effects, assuming the source is a delta function. In fitting the STF, a low-pass filter with a 1 MHz cutoff was applied to mitigate high-frequency artifacts ($\sim 1 \mu\text{s}$). Does this process limit the minimum resolvable source duration? This can be validated by applying the same data process to the theoretical Green's function to check whether there is a minimum resolvable duration in the first P-displacement pulse.

Thank you for the suggestion regarding the minimum resolvable source duration of the GP events. We performed a series of analyses to evaluate the limitations in resolving the source duration imposed by both the band-pass filtering and the attenuation correction, using synthetic Green's functions computed with the finite-difference software OpenSWPC.

We first examined the displacement waveforms of the raw Green's function and its band-pass-filtered counterpart. The Green's function was computed assuming the propagation path from P3 to receiver AS08, with a source-receiver distance of 185.2 mm. To approximate an impulsive source, the Green's function was convolved with a cosine-type STF with a duration of 1.0 μs . The seismic moment was arbitrarily set to 1.0 Nm. We then applied the same two-way third-order Butterworth band-pass filter (0.1–1 MHz) used in the main analysis to evaluate the P-wave displacement pulse width. Fig. R11 shows the raw and filtered Green's functions. We confirmed that the filtered P-wave pulse can be fit with a cosine STF with a duration of 1.6 μs . Therefore, the observed non-self-similar GP events, which have source durations of approximately 2.5 μs , are not limited by the band-pass filtering.

Fig. R11 Comparison of raw and band-pass-filtered Green's functions. The right panel shows a zoomed view of the P-wave displacement pulse.

We next investigated the resolution bias introduced by attenuation and its deconvolution. To mimic attenuation, we convolved the attenuation factor with the Green's function $g(\omega)$ as

$$u_1(\omega) = g(\omega)B(\omega),$$

where the attenuation factor is defined as

$$B(\omega) = \exp\left(-\frac{\omega t_p}{2Q(\omega)}\right),$$

with t_p is the P wave travel time and $Q(\omega)$ is the frequency-dependent quality factor estimated from the ball-drop experiments (See Method in the main text). In principle, perfect deconvolution of $B(\omega)$ would fully recover the source spectrum. However, in practice, a water-level is applied to stabilize the deconvolution, which limits the restoration of high-frequency components. To quantify this effect, we computed

$$u_2(\omega) = g(\omega)B(\omega)/B^{wlv}(\omega),$$

Where the stabilized attenuation factor is defined as

$$B^{wlv}(\omega) = \max\{|B(\omega)|, k|B(\omega)|_{max}\}.$$

Fig. R12 shows examples of the waveforms before and after the deconvolution with the water-level constraint. As source distance increases, the minimum resolvable source duration becomes longer due to the increased attenuation and the resulting water-level bias. However, the estimated resolution limits remain generally below $\sim 2.0 \mu\text{s}$, indicating that the observed non-self-similar source durations ($\sim 2.5 \mu\text{s}$) are above the method's resolution threshold and can therefore be reliably resolved.

Fig. R12 Attenuation effects for four AE sensors that limits the resolution of source duration. The black traces show the Green's functions after convolution with the attenuation factor, and the red traces show the results after deconvolution using the attenuation factor with a water-level parameter of $k = 0.3$, consistent with the main analysis.

3. The spectral ratio analysis can better constrain the relative seismic moment and relative source duration when fitted with a given source function model, such as the spectrum of the cosine STF. I suggest moving Fig. S7e to one of the main figures, as it provides additional constraints on the scaling between relative moment and duration. The authors state that the stacked spectral ratios from the four AEs are nearly flat, but I observe some slope differences that may indicate variations in source duration. The relationship between relative moment and relative duration, if derived from the spectral analysis, could be compared with the scaling shown in Fig. 3c, which I think would enhance the conclusion of this paper.

Thank you for the recommendation to include the spectral ratio analysis in the main figure. This addition helps clarify non-self-similar scaling while suggesting that subtle variations in source durations exist. We modified Fig. 3 by adding the stacked spectral ratio as shown in Fig. R13.

Fig. R13 Modified version of Fig 3 with adding spectral ratio analysis. Panel (d) shows the stacked spectral ratio computed across the four AE sensors closest to the GP. The spectra are approximately flat over the analyzed frequency range, while a subtle systematic slope is observed, which is consistent with variations in source duration between larger and smaller GP events.

We have also expanded the discussion of the spectral ratio analysis in the revised manuscript as follows:

L249: Another metric to evaluate the detailed scaling of non-self-similar events is the ratio of their source spectra (e.g., Lengliné et al., 2014; Cauchie et al., 2020; Nakajima & Hasegawa, 2024). We conducted spectral ratio analysis for five representative non-self-similar GP events (D24, D50, D52, D72, and D129). Raw AE waveforms were used without correcting for instrumental response, sensor coupling, or attenuation, as these effects are expected to cancel in the spectral ratio (Note S1). The amplitude spectra for the recorded waveforms are shown in Fig. S8. The stacked spectral ratios from four AE sensors closest to P3 are shown in Fig. 3d. The nearly flat spectral ratios indicate only minor differences in corner frequency, whereas the decreasing trend observed particularly for D24 is consistent with the weak scaling behavior inferred from Fig. 3c. Although the detailed scaling remains somewhat sensitive to the choice of fitting bandwidth and window functions, the spectral ratio results are compatible with the non-self-similar behavior identified in the time-domain analysis.

4. In the dynamic rupture simulations, the initial shear stresses are prescribed by trial and error, assuming that the entire gouge patch is ruptured. I am curious whether the strain gauges and gap sensors installed on the side surface could provide independent static constraints for the dynamic rupture model. Otherwise, the dynamic simulation results offer relatively limited insights in the current form.

We examined the coseismic shear-stress changes recorded by the strain gauges, but no clear signal was detected, as shown in Fig. R14. This is most likely because the stress change associated with a rupture on the gouge patch is too small to be resolved at rather far locations of the side wall. The gap sensor also does not have sufficient resolution to capture coseismic slip due to the small signal amplitude and limitations in its sensitivity. We therefore recognize the limited ability of these measurements to constrain the dynamic rupture model, particularly regarding the extent of the source region and the slip distribution.

Recent work by Dublanche et al. (2024) performed kinematic slip inversion using strain gauges installed on the side surface of a fault. Their synthetic tests showed that the inversion becomes unstable unless sensors are located inside the fault. This suggests that improving constraints for dynamic rupture modeling likely requires approaches beyond simply increasing the sensitivity or number of side-surface sensors.

We added the text in the manuscript as follows:

L293: The final rupture size can, in principle, be constrained through kinematic inversion of static strain or slip changes (e.g., Dublanche et al., 2024). However, no analyzable static offsets associated with the GP events are detected in either the strain or slip records, likely because the signals fall below the resolution limit of the measurements. While the final rupture extent remains unconstrained by static inversion, we assume that the source patch that produces positive stress

drop coincides with the GP dimension, and that the final rupture extent is governed by the rupture dynamics under the prescribed barrier conditions.

We also revised the section of dynamic rupture modeling in response to the feedback from General Comment #1 of Reviewer #1 to clarify that the proposed model is not fully constrained, but was intended to propose a feasible model, which potentially explain the factors contributing the non-self-similar scaling (L272-285).

Fig. R14 Shear stress change during **a**, stick–slip event and **b**, zoomed window between 40 ms and 50 ms. Black lines show the shear stress change measured by strain gauges along the fault. The red dashed line marks the onset timing of the GP event (D129, the largest target event), and the red circle indicates the location of the associated GP.

Perhaps the authors could also calculate the energy dissipation to discuss the potential fracture energy scaling (Kammer et al., 2024), comparing non-self-similar and self-similar scenarios.

As a first step, we assessed the fracture energy scaling for the non-self-similar events analyzed in this study by using the parameters adopted in the best-fit dynamic rupture models for the target events (Fig. R15). The corresponding values of G_{IIc} [J/m^2] are summarized in Table S2. The resulting scaling is broadly consistent with trends reported in previous studies, although the inferred scaling exponent is somewhat smaller than that typically obtained in laboratory experiments. While this topic lies outside the focus of the present manuscript, we are interested in investigating how the scaling may diverge between non-self-similar and self-similar events in future work.

Fig. R15 Scaling of fracture energy for the best-fit dynamic rupture models of the non-self-similar events. Black dots represent the dataset compiled by Cocco et al. (2023), for which we plot only the seismological observations. Red squares indicate the fracture energy assigned to the gouge patch for the five target events.

5. Natural faults may exhibit asperities of varying length scales. Under the condition of fixed asperity size, source duration may increase weakly with seismic moment, as proposed in this study. However, if data with diverse asperity sizes are combined, bridging laboratory and natural earthquake observations, could a self-similar scaling or other scaling emerge?

Thank you for the insightful comment. We consider a conceptual framework that faults host asperities with various sizes, each associated with the characteristic time scale controlled by its dimension. When events originating from asperities of a wide range of sizes, from laboratory to natural scales, are considered together, the overall behavior would follow self-similar scaling, consistent with the prediction by classical shear-crack models under approximately constant stress drop.

In contrast, for events associated with a given asperity size, variations in seismic moment can occur while the characteristic timescale remains nearly constant, leading to non-self-similar scaling as investigated in this study. This conceptual scaling is illustrated in Fig. R16.

This interpretation remains prospective. Further investigation will require theoretical development, laboratory experiments using gouge patches of different sizes, and analysis of natural earthquake observations.

We have added Fig. R16 as Supplementary Fig. S14 and incorporated this perspective into the Discussion section as follows:

L523: Natural observations of non-self-similar earthquakes can provide valuable perspective for clarifying the overall scaling laws of earthquakes bridging laboratory and natural fault systems. A conceptual framework is that the characteristic time scales are governed by asperity size, such that

fault systems hosting asperities of various dimensions collectively exhibit self-similar scaling, as predicted by classical shear-crack models. In contrast, for events associated with a given asperity size, variations in seismic moment can occur while the characteristic timescale remains nearly constant, leading to non-self-similar scaling as investigated in this study. This conceptual scaling, encompassing both self-similar and non-self-similar behaviors, is illustrated in Fig. S14.

Fig. R16 Schematic illustration of potential scaling behavior for different patch sizes. The thick black line represents the self-similar scaling $T_w \sim M_0^{1/3}$ for different patch sizes. Thin blue lines indicate non-self-similar scaling branches controlled by the patch size.

In correspondence with this revision, we rewrote a paragraph in the Discussion as follows:

L460: The proposed model, which assumes a fixed source patch size with variable stress drop, indicates that self-healing friction can effectively shorten the source duration of larger events on a fixed source patch, even in the absence of a strong rupture barrier. As a consequence, this model complements existing frameworks and broadens the range of tectonic settings under which non-self-similar earthquakes may occur. Moreover, the GP events provide insights into the earthquake generation processes not only on tectonic faults but also in volcanic environments (e.g., Hotovec-Ellis et al., 2022) and at subglacial asperities on ice–bedrock interfaces (Zoet et al., 2012).

Minor comments:

1. Lines 15-16: Is it possible to validate if the average rupture velocity scales with the moment from dynamic rupture simulations?

We assessed the evolution of rupture velocity in the dynamic rupture simulations with different seismic moment for the target GP events (Fig. R17). Given that the seismic ratio (e.g., Andrews, 1976) is very small ($\ll 1$) for all the events in our preferred models, the rupture front rapidly accelerates toward supershear speed. The smallest event, D24, reaches a velocity slightly above the sub-Rayleigh regime, whereas the larger events (D52 and D129) transition to fully developed supershear propagation.

Although this behavior should be interpreted with caution, as the dynamic simulations are not tightly constrained by direct observations such as kinematic inversions, the results indicate that the rupture velocity is largely insensitive to differences in seismic moment in our proposed models.

Fig. R17 Space–time evolution of slip velocity for the three target events. Panels a–c show dynamic rupture models with different seismic moment of target events: **a**, D24 ($M_0=0.07$ Nm), **b**, D52 ($M_0=0.47$ Nm), **c**, D129 ($M_0=1.15$ Nm). The slip-velocity field is extracted along the cross-section through the patch center ($z = 0$). Colors indicate slip velocity, and the red dashed line denotes the reference slope corresponding to the Rayleigh-wave speed. The gray zone corresponds to the patch region.

2. Line 19: The size and shape of the fault asperity are controlled, but whether the earthquake sources have fixed size and shape remains to be verified.

We agree that the actual ruptured area associated with the gouge patch events is not directly constrained, for example by kinematic inversion or other near-fault measurements, as discussed throughout this rebuttal letter. We have therefore revised the manuscript to more clearly articulate the scope of the proposed dynamic rupture model (see our response to General Comment #1 of Reviewer #1). Rather than presenting it as the most definitive explanation, we now clarify that it represents one plausible model among several possible scenarios.

3. Fig. 2a: Why are the amplitudes of the foreshock waveforms (the one before ID72) larger than those of the mainshock?

The amplitudes appear larger for the foreshock in Fig. 2a because the mainshock signals reach the ± 10 V dynamic range of the AE sensors, resulting in clipping. These clipped signals are subsequently mitigated by band-pass filtering between 0.1 and 1 MHz, which reduces the apparent amplitudes of the mainshock waveforms. When the data are plotted without this filter, the mainshock also exhibits comparably large amplitudes. An unfiltered version of Fig. 2a is provided as Fig. R18 in this rebuttal letter.

To clarify this point, we added the following explanation to the caption of Fig. 2a:

Caption of Fig. 2: Large main-shock amplitudes are present in the raw data but are clipped in the recording and do not appear after band-pass filtering

Fig. R18 Replot of Fig. 2a without applying band-pass filter to the AE waveforms. The mainshock signal is clipped but exhibits amplitudes comparable to those of the foreshocks.

4. Line 43: It is ambiguous whether “This type of scaling” refer to “the self-similar scaling” or “the non-self-similar scaling”, as the latter has not been mentioned before this paragraph. I suggest revising it “The non-self-similar scaling”.

Thank you for the clarification. We have revised the text accordingly, as follows:

L50: This type of scaling This scaling relationship, referred to as non-self-similar scaling, was first reported in earthquake clusters at Parkfield by Harrington and Brodsky (2009)

5. Lines 39-40: It would be better to cite relevant literature here.

We updated the text as follows:

L46: However, deviations from this scaling have been observed in some earthquake clusters, showing a nearly constant source duration regardless of variations in seismic moment (e.g., Harrington & Brodsky, 2009; Lin et al., 2016).

6. Lines 45-47: Consider including the result of this paper, such as “Nakajima and Hasegawa (2024) compiled the observations that ..., showing a non-self-similar scaling.”?

Yes. We have revised the text accordingly as follows:

L53: Nakajima & Hasegawa (2024) compiled observations of repeating earthquakes that exhibit non-self-similar scaling across various tectonic settings, including the continental crust, intraslab regions, and plate boundaries beneath the Japanese Islands.

7. Line 51: “reproduce” -> “explain”

Thank you for the suggestion. We have revised the wording accordingly at L77.

8. Line 57: “While the previous studies” -> “While previous studies”

The text has been updated as suggested at L83.

9. Lines 156-158: Fitting the spectral ratio may provide better constraints on relative seismic moment and source duration for the scaling, as the effects of unknown Green’s function and instrument response can be well cancelled out by the spectral ratio analysis.

We agree that spectral ratio analysis provides robust constraints on relative source parameters. Although the retrieval of relative source parameters from spectral ratios of GP events is somewhat challenging due to spectral instability, we expanded the discussion of the spectral ratio analysis in the revised manuscript and moved the relevant figure to Fig. 3d to demonstrate its consistency with the time-domain analysis (see response to Major Comment #3).

10. Line 406: shall it be “the input (LDV)” or “the output (LDV)”?

We confirm that “the input (LDV)” is consistent with our formulation. The transfer function $H(\omega)$ is defined as

$$H(\omega) = \frac{Y(\omega)}{U(\omega)}$$

where $Y(\omega)$ is the output spectrum recorded by the AE sensor and $U(\omega)$ is the input vibration signal inferred from the LDV measurements. Accordingly, the Bode plot shown in Fig. S19a (formerly Fig. S14a in the original version) correctly represents the system response obtained by dividing the AE sensor spectrum by the LDV-inferred input spectrum.

We have revised the text as follows:

L610: The amplitude and frequency response of the AE sensor, obtained by comparing the output spectrum of the AE sensor with the input vibration signal inferred from the LDV measurements, are well reproduced by the TF with the optimized poles and zeros.

11. Line 419-421: In evaluating the sensor coupling factors, the ball-drop test was performed from a height of 0.5 m. Why not test different heights? Since this paper focuses on the scaling of different magnitude, it may be helpful to test the effects of different magnitudes of the ball-drop impacts.

The ball-drop tests were conducted primarily to evaluate sensor coupling factors, rather than to calibrate the intrinsic sensor gain, which was independently calibrated using piezoelectric transducers that generate more stable source pulses.

In addition, the linearity of the sensor output amplitude was assessed by performing ball-drop tests at different source–receiver distances, as shown in Fig. S20, which allowed us to span a range of input amplitudes without varying the drop height.

We acknowledge that testing different drop heights would further extend the calibration range toward higher amplitudes, which would be useful for future experiments involving larger gouge patches expected to generate stronger signals.

12. Supplementary Note S1: For better readability, I suggest adding the complete name of GP at the beginning of the note, although it has been mentioned in the main text.

Thank you for the suggestion. We have updated Supplementary Note S1 to spell out “gouge patch (GP)” at the beginning for clarity.

Addendum

We have revised the labeling of the GP events to adopt a unified notation with the prefix “D” (e.g., D129), replacing previously mixed use of labels such as M129 and D129. The original labeling scheme was intended to distinguish parameters associated with modeling and observations, but it introduced unnecessary complexity into the manuscript. Throughout the revised manuscript and this rebuttal letter, we consistently refer to the target events using unified event-number notation, with appropriate contextual explanations provided where needed, to improve clarity and avoid confusion. The annotations of event labels in Figs. 2-5 are accordingly updated. As part of this revision, Fig. S5 was also updated by removing the foreshock/aftershock suffixes (e.g., 129F, 129A), and the corresponding information is now summarized in Table S1.

In addition, we applied minor figure updates to improve the clarity of ball-drop calibration figures. These changes affect Figs. R19 and R20 and are documented below for completeness.

Fig. R19 Modified version of Fig. S20 (formerly Fig. S15 in the original version) with added annotations. Labels for ball-drop sources (a-d) and a sensor annotation (AS18) were added. These annotations are consistent with the ball-drop waveform comparison shown in Fig. S23 (formerly Fig. S18) and have been incorporated into Fig. S20 (formerly Fig. S15).

Fig. R20 Modified version of Fig. S23 (formerly Fig. S18 in the original version) with added panel labels. Panel labels (a-h) were added to the ball-drop waveform comparison, and the caption was updated accordingly.

References for the Rebuttal Letter

Andrews, D. J. (1976) Rupture velocity of plane strain shear cracks. *J. Geophys. Res.*, 81(32):5679--5687, <https://doi.org/10.1029/JB081i032p05679>

Dunham, E. M. and Bhat, H. S. (2008) Attenuation of radiated ground motion and stresses from three-dimensional supershear ruptures. *J. Geophys. Res.*, 113, B08319, <https://doi.org/10.1029/2007JB005182>

Palmer, A. C. and Rice, J. R. (1973) Growth of slip surfaces in progressive failure of over-consolidated clay. *Proc. R. Soc. Lond. Ser-A*, 332:527--548, <https://doi.org/10.1098/rspa.1973.0040>

Tsai, V. C., & Hirth, G. (2020). Elastic impact consequences for high-frequency earthquake ground motion. *Geophys. Res. Lett.*, 47(5), e2019GL086302. <https://doi.org/10.1029/2019GL086302>

Response to Reviewers (Second Round): Dynamics of non-self-similar earthquakes illuminated by a controlled fault asperity

Kurama Okubo, Futoshi Yamashita, Eiichi Fukuyama
(NCOMMS-25-32014A)

March 19, 2026

We again thank the Editor, the Associate Editor, and the reviewers for their feedback on the revised manuscript. Below, we provide responses to the additional comments with the corresponding updates to the manuscript. Black text indicates the original reviewer comments, blue text represents our responses, and red text shows the revised text included in the manuscript. Line numbers in the revised text correspond to those in the version with track changes.

Comments by Reviewer #1

All the details of the code are fine. I was just wondering because the authors mentioned a modified version of OpenSWPC to take into account the specific boundary condition of the experiment, is this modified version of this software available somewhere?

The modified version of OpenSWPC, which incorporates free-surface conditions along the boundaries of the rock specimen, is maintained in the GitHub repository associated with this study. A minimum working example notebook demonstrating the simulation setup and results is also available at the following link:

https://github.com/kura-okubo/4mNonSelfSim_OpenSWPC/blob/develop/example_balldrop/example_balldrop_result.ipynb

We have revised the Code Availability section in the main text as follows:

L709: The numerical simulations of waveform propagation were performed using an extended version of OpenSWPC v5.1.0 (Maeda et al., 2017), modified to implement free-surface conditions along the boundaries of the rock specimen. The code is maintained in the GitHub repository 4mNonSelfSim_OpenSWPC and archived at <https://doi.org/10.5281/zenodo.15288446>.

Comments by Reviewer #3

The authors have appropriately addressed most reviewer concerns, except for the two following points.

First, if the minimum resolvable source duration is approximately 2.0 μs , the authors should justify whether the truncated durations of events D118 and D126 affect the inferred scaling. If the true durations of these two events are allowed to be shorter, the scaling slope would be expected to exceed 0.04.

To avoid potential artifacts due to the truncated durations of D118 and D126, these events were already excluded from the linear regression analysis in the original manuscript. We acknowledge

that the inferred scaling slope could be larger if the true source durations of these events were measurable without this limitation. Future experiments with larger gouge patches, which would allow a wider range of source durations to fall within the measurable bandwidth, will help further constrain the overall scaling relationship associated with the non-self-similar cluster.

We have clarified this point in the manuscript as follows:

L207: For clarity, we define the non-self-similar cluster as all GP events shown in Fig. 3c, excluding the two short-duration events, D118 and D126, whose durations approach the minimum resolvable limit of the measurement and may therefore bias the scaling analysis. We performed a linear regression on this cluster, yielding a scaling exponent of 0.04 ± 0.02 .

Second, the manuscript uses “dynamic source model” for the experiments and “dynamic rupture model” for the simulations, which is somewhat confusing. It is more common to use dynamic rupture model to refer to numerical simulations. But it is not a major issue.

Thank you for the comment. The term “dynamic source model” appeared only in the abstract, and we have revised it to “dynamic rupture model” to avoid potential confusion.

Review of Manuscript: "Dynamics of non-self-similar earthquakes illuminated by a controlled fault asperity"

This manuscript presents interesting results from laboratory earthquake experiments with seven 8 mm-diameter circular gouge patches placed on a meter-scale fault interface. The authors propose that the laboratory earthquakes, nucleated within the gouge patches, show a non-self-similar scaling, i.e., the source duration weakly dependent on the seismic moment. The non-self-similar scaling was previously reported in natural faults. They use dynamic rupture simulations to reproduce the observed non-self-similarity, and suggest that a combination of variable stress drop and self-healing friction is needed to fit the observations, with the latter preventing rupture extending outside the gouge patch. The laboratory experiments in this study provide interesting new data that can help elucidate the mechanisms of small earthquakes. And these data may connect the scaling relation across different length scales from laboratory observations to larger natural earthquakes. I have several comments about this manuscript before it can be published.

Major comments:

1. Did all the laboratory earthquakes rupture the entire gouge patch? If the smallest events only ruptured a small fraction of the gouge patch, can the current measurements detect and constrain them? It seems that the dynamic rupture simulations have implicitly assumed the condition of a fixed rupture size. I appreciate the authors' efforts to account for the attenuation effects and the precise calibration of the instruments to reduce possible biases, but I think a missing element is verifying whether the instruments can constrain the source duration of smaller events when their durations are much shorter than 2 μs (the smallest event). Although Figure S17 has shown the ball-drop impact, its duration of 5 μs (found in Fig S17) is longer than that of the largest laboratory earthquake ($\sim 3 \mu\text{s}$), and therefore it does not directly verify the resolution for short source durations.
2. It may be helpful to show the Green's function of each AE with the attenuation effects, assuming the source is a delta function. In fitting the STF, a low-pass filter with a 1 MHz cutoff was applied to mitigate high-frequency artifacts ($\sim 1 \mu\text{s}$). Does this process limit the minimum resolvable source duration? This can be validated by applying the same data process to the theoretical Green's function to check whether there is a minimum resolvable duration in the first P-displacement pulse.

3. The spectral ratio analysis can better constrain the relative seismic moment and relative source duration when fitted with a given source function model, such as the spectrum of the cosine STF. I suggest moving Fig. S7e to one of the main figures, as it provides additional constraints on the scaling between relative moment and duration. The authors state that the stacked spectral ratios from the four AEs are nearly flat, but I observe some slope differences that may indicate variations in source duration. The relationship between relative moment and relative duration, if derived from the spectral analysis, could be compared with the scaling shown in Fig. 3c, which I think would enhance the conclusion of this paper.
4. In the dynamic rupture simulations, the initial shear stresses are prescribed by trial and error, assuming that the entire gouge patch is ruptured. I am curious whether the strain gauges and gap sensors installed on the side surface could provide independent static constraints for the dynamic rupture model. Otherwise, the dynamic simulation results offer relatively limited insights in the current form. Perhaps the authors could also calculate the energy dissipation to discuss the potential fracture energy scaling (Kammer et al., 2024), comparing non-self-similar and self-similar scenarios.
5. Natural faults may exhibit asperities of varying length scales. Under the condition of fixed asperity size, source duration may increase weakly with seismic moment, as proposed in this study. However, if data with diverse asperity sizes are combined, bridging laboratory and natural earthquake observations, could a self-similar scaling or other scaling emerge?

Minor comments:

1. Lines 15-16: Is it possible to validate if the average rupture velocity scales with the moment from dynamic rupture simulations?
2. Line 19: The size and shape of the fault asperity are controlled, but whether the earthquake sources have fixed size and shape remains to be verified.
3. Fig. 2a: Why are the amplitudes of the foreshock waveforms (the one before ID72) larger than those of the mainshock?
4. Line 43: It is ambiguous whether “This type of scaling” refer to “the self-similar scaling” or “the non-self-similar scaling”, as the latter has not been mentioned before this paragraph. I suggest revising it “The non-self-similar scaling”.
5. Lines 39-40: It would be better to cite relevant literature here.

6. Lines 45-47: Consider including the result of this paper, such as “Nakajima and Hasegawa (2024) compiled the observations that ..., showing a non-self-similar scaling.”?
7. Line 51: “reproduce” -> “explain”
8. Line 57: “While the previous studies” -> “While previous studies”
9. Lines 156-158: Fitting the spectral ratio may provide better constraints on relative seismic moment and source duration for the scaling, as the effects of unknown Green’s function and instrument response can be well cancelled out by the spectral ratio analysis.
10. Line 406: shall it be “the input (LDV)” or “the output (LDV)”?
11. Line 419-421: In evaluating the sensor coupling factors, the ball-drop test was performed from a height of 0.5 m. Why not test different heights? Since this paper focuses on the scaling of different magnitude, it may be helpful to test the effects of different magnitudes of the ball-drop impacts.
12. Supplementary Note S1: For better readability, I suggest adding the complete name of GP at the beginning of the note, although it has been mentioned in the main text.

Reference:

Kammer, D. S., G. C. McLaskey, R. E. Abercrombie, J. P. Ampuero, C. Cattania, M. Cocco, L. Dal Zilio, G. Dresen, A. A. Gabriel, C. Y. Ke, C. Marone, P. A. Selvadurai and E. Tinti (2024). "Earthquake energy dissipation in a fracture mechanics framework." *Nat Commun* 15(1): 4736.